# The default network dominates neural responses to evolving movie stories

Enning Yang [1,2], Filip Milisav [1], Jakub Kopal [1,2], Avram J. Holmes [3], Georgios D. Mitsis[4], Bratislav Misic [1], Emily S. Finn [5] & Danilo Bzdok [1,2] ✉

Neuroscientific studies exploring real-world dynamic perception often overlook the influence of continuous changes in narrative content. In our research, we utilize machine learning tools for natural language processing to examine the relationship between movie narratives and neural responses. By analyzing over 50,000 brain images of participants watching Forrest Gump from the studyforrest dataset, we find distinct brain states that capture unique semantic aspects of the unfolding story. The default network, associated with semantic information integration, is the most engaged during movie watching. Furthermore, we identify two mechanisms that underlie how the default network liaises with the amygdala and hippocampus. Our findings demonstrate effective approaches to understanding neural processes in everyday situations and their relation to conscious awareness.

The imaging neuroscience community has recently embarked on a new wave of experiments—naturalistic neuroscience. This "third wave paradigm"[1] endorses videos, spoken narratives, and other real-life sensory presentations, aiming for fuller understanding of the brain mechanisms that realize the processing of dynamic, ecologically valid stimuli[2]. Such naturalistic studies promise to better emulate the complex perception and behavior of everyday life. The more realistic experimental outlets unlocked insights into some classes of neuro-cognitive processes that may take us closer to real-world cognition[3].

In fact, widely adopted content-free resting state experiments have a critical drawback: self-generated random thoughts lie outside of the reach of experimental control. Instead, in the naturalistic settings, the movie-induced brain dynamics are synchronized across different subjects by virtue of watching the same full movie i.e., time-locked stimulus[4]. New methods need to be deployed to take full advantage of time-locked neural responses across subjects[5], thus improving the predictability of trait-like phenotypes[6,7]. In sum, studies benefitting from naturalistic stimuli have paved the way for unprecedented findings in various areas, such as the temporal characteristics across the neural processing hierarchy[8].

However, to be able to zoom in on the contextual richness that evokes brain dynamics in human experience, we need to take on new methodological challenges due to the amount, complexity, and continuous quality of naturalistic stimulation[2]. Many such previous studies aimed to develop new approaches to extract different aspects of stimulus-driven brain dynamics. In these studies, traditional cognitive concepts or human-crafted naturalistic features were commonly used to reveal and explain findings on brain dynamics[9,10]. An emerging line of research started to adopt machine learning-based naturalistic stimulus representations to explain the neural activity signals[11,12]. Such efforts have typically focused on low-level visual representations or mid-level categorical processing tasks. Hence, the analysis of high-level contextual information and heteromodal information integration processes subserved by the higher association cortex remains under-investigated. Natural language processing (NLP) tools respond especially well to this need to distill high-level semantic features of the environment. As a core motivation for the present endeavor, we argue that NLP is a promising lever to explore higher-order functions in different networks along the brain's processing hierarchy, especially its deeper associative brain network layers, far removed from sensory input processing.

[1]Department of Biomedical Engineering, TheNeuro—Montreal Neurological Institute (MNI), McConnell Brain Imaging Centre (BIC), McGill University, Montreal, QC, Canada. [2]Mila—Quebec Artificial Intelligence Institute, Montreal, QC, Canada. [3]Department of Psychology and Psychiatry, Yale University, New Haven, CT, USA. [4]Department of Bioengineering, McGill University, Montreal, QC, Canada. [5]Department of Psychological and Brain Sciences, Dartmouth College, Hanover, NH, USA. ✉e-mail: danilo.bzdok@mcgill.ca

NLP has become an increasingly valuable tool for studying the human language systems implemented in the human brain. Several studies have used advanced NLP techniques to build association maps between language features and brain activity. The diverse set of experimental tasks include predicting movie scenes[13], detecting semantic selectivity[14], and decoding semantic content[15,16]. In particular, the usage of NLP techniques to access semantic features of human language has been shown to be successfully mappable to brain activity before[17,18]. The advent of large language models, such as enabled by transformer architectures, has pointed to exciting features of the human brain, such as the possibility of shared computational design principles in form of next-word prediction mechanisms[19–21]. Additionally, large open neuroscience datasets have emerged as a promising way to accelerate research in this area[22]. These developments are likely to continue to advance new insights into the relationship between language processing and neural processing systems.

Over the last few years, the machine learning community has seen progress in several areas as NLP technologies have rapidly matured[23]. Some state-of-the-art NLP models contain billions of parameters, which may outnumber the >80 million neurons of the human brain[24,25]. Text generated from large NLP model architectures has been reminiscent of some aspects of human conscious awareness[26–28]. By extracting and integrating the semantic structure drawn directly from human language itself, NLP-based analyses have started to extend interpretations of experimentally induced changes in conscious awareness and their corresponding brain representations[29]. Such usage of natural human language analyses for scientific discovery encourages to rethink the reliability and measurability of some traditional cognitive concepts: Is the brain organized according to the psychological definitions that neuroscientists have inherited from the behavioral sciences?[30] It is hard to scrutinize, to what extent, long-standing cognitive notions, like "valence", "fear", or "arousal", bear clear-cut instantiations in human brain function. As an attractive alternative strategy, our study will embrace the stature of human language itself, which we humans use effectively to describe and understand our daily reality, to contextualize complex neural activity responses observed during movie engagement.

Narrow concepts and basic emotions, such as fear, may not afford a sufficiently rich description of many sophisticated neural processes—especially those animated by the higher association cortex that are particularly well developed in humans[31]. More nuanced semantic descriptions are better posed to help decipher the neural computations of the deepest association layers of the human network hierarchy. Many of today's experimental paradigms used for studying higher-order information processing tend to hinge on vague definitions[32–34]. NLP here now offers the potential of disclosing complex aspects of semantic structure in the movie material[35]. Therefore, the strategy of porting tools from NLP to imaging neuroscience may allow to more cleanly disentangle semantically denotable higher-order brain functions in humans against the brain's housekeeping signals and background noise. In sum, the high-level brain processes, underlying real-world cognition, may not be adequately describable by low-level concepts, like fear. Hence, harnessing the power of emerging NLP technologies can build a bridge that begins to brain dynamics and the granular semantics of human natural language.

For these reasons, we brought to bear NLP techniques to mine brain-imaging experiments administering a 2-h movie. We could thus peel apart how brain network layers differentially tie into dynamic context information in response to the movie narrative. Taking advantage of the studyforrest resource's >50,000 timepoints, we could train separate hidden Markov models for each of 15 subjects to chart limbic-neocortical region-network combinations. Using the seven networks from the Schaefer-Yeo atlas[36], we tried to cover the brain networks spanning from lower, most sensory, unimodal layers to the highest, most sensory-independent layers of neural processing. In this way, we directly compared canonical brain network layers in tracking salient movie events. We also explicitly linked the seven neocortical networks with two key partners of the medial-temporal sub-neocortical system—the amygdala and the hippocampus, given recent updates in anatomical understanding[37]. Taken together, our collection of derived brain states at subregion resolution offers detailed views on subject-level differences, limbic-neocortical coupling regimes, and specific roles for different brain network layers during naturalistic stimulation. We integrate external information from curated human annotations and derived elements of the evolving movie narrative. In so doing, our discovered brain dynamic signatures are interpreted by traditional concepts and our (timepoint-level) semantic underpinnings.

To foreshadow our key contributions, we have developed an analytical framework enabling hidden Markov models (HMMs) at the single-subject level to analyze the dynamic functional connectivity in the brain during naturalistic movie watching. This approach allowed us to understand the idiosyncrasies of each individual's responses to the movie material and provides a comprehensive and faithful picture of the individual's functional coupling dynamics evoked by naturalistic stimulation. Furthermore, we have used unprecedented anatomical granularity at the subregion-level to chart movie responses, mapping out neural responses in 18 amygdala subregions, and in 38 hippocampal subregions. The proposed approach allowed us to build on evidence from biological pathways that have been previously reported in invasive animal experiments. Additionally, we have compared the value of human-made semantic labels and data-driven semantic labels in providing insight into neural responses during movie watching, showing the relevance of data-driven labels to specific movie events and highlighting the broader role of the default network (DN) in pooling and binding brain-wide information.

## Results
### Inferring brain state probabilities of 14 different limbic–neocortical combinations
Leveraging the wealth of >50,000 brain scanning timepoints, 3543 for each of the 15 subjects, we could train HMM solutions on a single-subject basis. We aimed to identify cliques of functional coupling partners between a given neocortical network and either hippocampus (HC) or amygdala (AM) subregions. In this way, we jointly analyzed data on neural activity responses from both neocortical and limbic subregions. First, we extracted subregion-wise averages of voxel brain activity responses to naturalistic movie stimulation based on the Schaefer-Yeo anatomical reference atlas at 100 subregion resolution[38]. We then performed a microanatomical segmentation (using Free-Surfer) on the structural brain anatomy (T1 brain scans) to delineate the 3D shapes of 38 HC and 18 AM subregions specific to each subject (two corresponding ones for each hemisphere, such as left and right CA1). Based on segmentation of the HC and AM at subregion granularity, we extracted neural activity time courses to supplement those of the neocortical networks. Henceforth, we refer to the amygdala and hippocampus as limbic, non-neocortical or subcortical structures, as they are not included in our cortical Schaefer-Yeo atlas. However, it should be noted that the hippocampus is part of the allocortex and therefore sometimes considered as subcortical and sometimes as a cortical region.

We estimated several HMM solutions in parallel, each fitting one specific combination of one neocortical network and one limbic region. As such, in each of the 15 subjects, we trained 14 HMMs (7 neocortical networks x 2 limbic subregion sets) from both limbic and neocortical neural activity, that is, 14 "region-network combinations" in total. Each dynamic model estimated subject-specific brain states along with their state probabilities along the movie (continuous state presence probabilities). Estimating four brain states per HMM was found to be a useful choice based on four distinct criteria of optimality

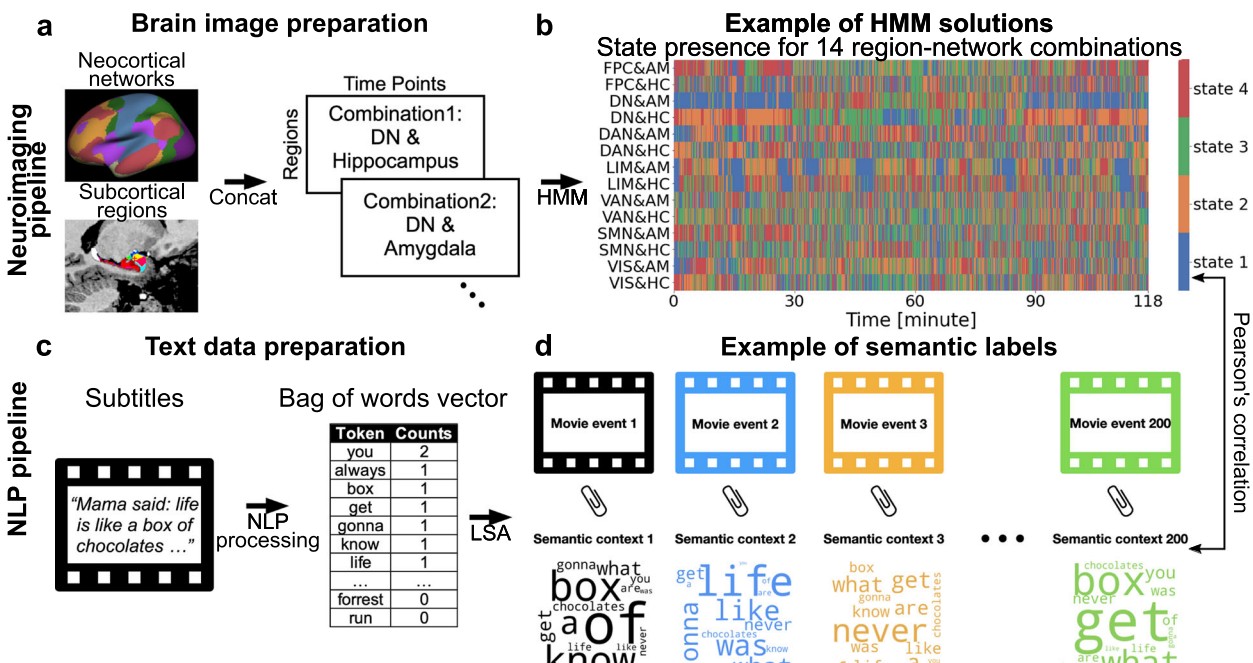

Fig. 1 | Quantitative analysis workflow. a As part of the brain-imaging data processing pipeline, the Schaefer-Yeo reference atlas served to extract neural responses during movie watching from 100 anatomical subregions from seven established functional brain networks, spanning from lower sensory to highly associative circuits. We supplemented the neocortical functional networks with neural activity from hippocampal and amygdalar subregion sets (summarized here as 'limbic regions') using rigorous microanatomical segmentation. Subsequently, the time series of 14 limbic-neocortical combinations provided the basis for estimating 14 separate hidden Markov models (HMM), for each subject. b The extracted state presence (color coding is specific to each region-network combination [row]) delineates the temporal dynamics of functional brain coupling cliques both vertically (across region-network combinations) and horizontally (across narrative shifts). c As part of the text data processing pipeline, movie subtitles and narrator descriptions served as raw text information. By means of natural language processing (NLP) techniques, the text was re-expressed as word occurrences (i.e., vocabulary word occurrence matrix). d The per-timepoint expressions of 200 semantic contexts were obtained from latent semantic analysis (LSA; cf. methods). Each color represents a unique semantic context, whose presence was inferred as proxy trajectories for underlying movie events. Short names for Schaefer-Yeo networks: VIS Visual network, SMN somatomotor network, DAN dorsal attention network, DN default network, LIM limbic network, VAN salience and ventral attention network, FPC Frontoparietal network. Source data are provided as a Source Data file.

(Supplementary Fig. 1). Additionally, we demonstrate the preference of seven cortical networks to the correlation criteria separately to eliminate potential biases (Supplementary Fig. 12). Therefore, our analytical framework was carefully tailored to capture rich information in the full-length 2-h movie. Together, the dynamic structured time-series modeling approach directly quantified subject- and subregion-level properties.

As an illustration of discernable results from a representative subject, we quantify the state presence of each of 14 region-network combinations for subject 1 (Fig. 1B). We found that the duration of continuous, uninterrupted occupation in each dynamic brain state (i.e., dwell time) showed distributions that were specific to each of the 14 region-network combinations. This model-derived quantity exposed the neural processing timescales of the event structures detected by a dynamic HMM. We thus tested how timescales of dynamic state events varied across different layers of the unimodal-to-associative neural processing hierarchy (Mesulam[39]). The lower-level canonical brain systems, such as the visual network (VIS), tended to be subject to faster locking in and out of a dynamic brain state[40], since they showed shorter dwell times and lower variance (e.g., VIS&HC: $\mu = 10.07$ s, $\sigma = 9.30$ s). Conversely, in the higher-level processing networks, brain states showed dwell times of longer duration and larger variance (e.g., DN&HC: $\mu = 25.58$ s, $\sigma = 33.20$ s). These observations suggested a divergence of naturalistic movie processing dynamics happening between lower-level networks versus higher-order networks.

We further examined the uncertainty of online activity of the dynamic brain states across the 2-h movie. To this end, we examined the variance of local average dwell times in the different segments of the movie to test if different parts of movie differ in brain processing timescales. The results on state volatility showed that dwell times pertaining to the DN (DN&HC: $\sigma = 6.30$ s; DN&AM: $\sigma = 5.23$ s) were much more volatile than in the other region-network combinations ($0.74$ s $< \sigma < 2.13$ s). These processing timescales from DN-tuned models were also more variable in subjects, while those of lower-level networks were more stable. These findings suggest that the unfolding movie narrative may be preferentially reflected in the state switching characteristics of the higher-order brain networks. This observation could be taken to suggest existing links between contextual movie information and particular brain states switching online and offline. Yet, the principled exploration of this potential link required the semantic dissection of the flow of the movie narrative.

### Semantic contexts were linked with brain states

We, therefore, investigated whether the movie-induced changes in functional coupling dynamics were associated with changes in the lattice of semantic contexts that together compose the plot (in continuous degrees). To examine the flow in contextual information, we carried out a quantitative dissection into 200 semantic context definitions that capture word usage trends with their shifts from moment to moment (Fig. 1d). The 200 discovered semantic contexts were extracted from the original subtitles of the movie by latent semantic analysis (LSA; Fig. 1c, d; cf. methods). The ensuing charting of semantic embedding trajectories yielded compact low-dimensional representations that track events along the continuous movie, which we integrated with the timestamps in the movie when a given semantic

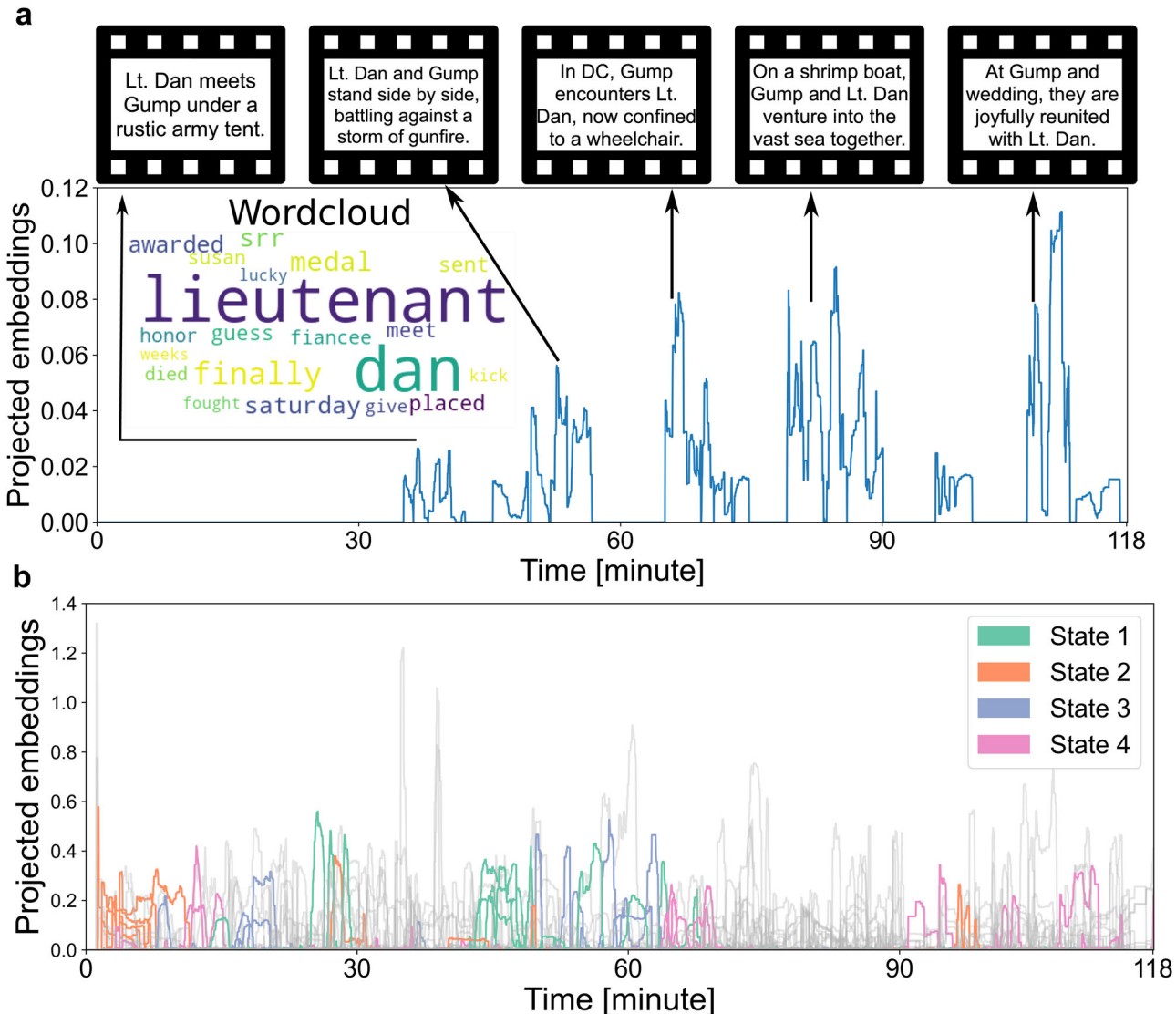

**Fig. 2 | Shifts in semantic contexts of the movie are tied to shifts in brain state dynamics.** We used NLP tools from natural language processing to decompose the movie subtitle corpus into 200 unique semantic contexts across thousands of vocabulary entries. The timepoint-wise quantification of semantic context occurrences offered the basis to link meaning with brain states. The contexts' associated wordclouds help interpret the semantics-brain links. **a** Middle left: The wordcloud for exemplary semantic context No. 103. The captured movie events centered around the character "Lieutenant Dan". Rest: This representative semantic context described recurring events for the same character (Lieutenant Dan), the associated movie clips (top row) were linked with the projected embedding peaks by an arrow, on which the text illustrated the brief movie events. **b** Subject hidden Markov model: each dynamic brain state (of subject 1's DN&AM model) was correlated with certain semantic contexts (colored according to brain state; see Supplementary Fig. 1 for choice of four states). The collection of remaining 160 contexts is shown in gray. As such, we performed a semantic dissection (via latent semantic analysis) of recurring movie themes into 200 unique semantic contexts, which related to complementary contextual information and distinct brain states. Source data are provided as a Source Data file.

context occurred. As a convenient synopsis for visualization, we used word clouds to summarize the most prominent word groups, as identified by our NLP framework. Many of the extracted semantic contexts showed expression peaks predominantly associated with a unique timestamp and therefore singled out a specific event. Yet, a variety of semantic contexts emerged to flag a theme that reoccurred in different parts of the movie, representing related events throughout the narrative. As one of many examples, the "Lieutenant Dan" component (Fig. 2a) indexed relevant events about the supporting character Lieutenant Dan across the 2-h movie. This semantic dimension showed high expressions with recurring peaks throughout the story (Fig. 2a) every time the character appeared in the movie. In so doing, we quantitatively dissected the semantic structure of the movie into small, abstract, and context-dependent expressions such that we could probe alignment of these features with the dynamic brain states.

To then interrogate associations of elements of the narrative with the brain states (semantics-brain links), we computed Pearson's correlations between the trajectory of state presence probabilities and the trajectory of semantic context expressions across the whole movie. We wished to describe how each unique brain state matches with the collection of semantic contexts detected in the movie. We thus identified each brain state's top 10 strongest semantics-brain links across 200 semantic contexts given a specific subject's HMM solutions of certain region-network combinations. We considered the top 10 links' average Pearson's correlation coefficients as an index for a state's general strength of linkage with the evolution of the movie plot. As an example, we illustrate subject 1's HMM results based on the DN&AM model (Fig. 2b). The obtained coefficients $r$ of association strengths were 0.20, 0.17, 0.12, and 0.18 (as measured by the average Pearson's correlation coefficient for the top 10 contexts) for state 1 to state 4,

respectively. Further, the sets of top semantics-brain links across brain states were shown to be mutually unrelated to each other. This observation supported that different brain states are tracking different facets of the narrative during the continuous movie stimulation. Taken together, our analyses delineated the strongest correspondences between the network functional coupling states and extracted movie story dimensions.

To complement the portfolio of automatically delineated semantic contexts and their corresponding brain manifestations, we analogously calculated Pearson's correlation coefficients to index possible associations between brain state expressions (presence probabilities) and 52 human-curated annotations (annotation-brain links) for each subject and region-network combination. This extensive collection of annotations, from post-hoc ratings, offered by the studyforrest resource covered a variety of concepts, commonly studied in the cognitive sciences, including emotions, salience, and key properties of the environment depicted in the movie (for full details see Supplementary Fig. 6). For example, regarding the DN&AM HMM solution, the top annotation-brain links were "Gump property" for state 1 (Pearson's correlation coefficient $r = 0.26$); "gratitude" for state 2 ($r = 0.08$); "Washington D.C." for state 3 ($r = 0.17$); and "Vietnam" for state 4 ($r = 0.29$). Taken together, these annotation-brain links showed the strongest associations with descriptions defined by humans in a top-down fashion, which complemented the purely data-driven semantics-brain links (cf. previous paragraph).

### Dynamic brain states track fine-grained contextual changes throughout the movie

The differentions between the semantic contexts and functional coupling states showed that we can characterize brain dynamics explicitly using rich movie information. Therefore, we next assessed the exhaustive collection of results of the multiple links between brain states and movie context features. Our analytical framework enabled us to explain the subregion-level brain signatures and the functional coupling dynamics with external descriptions of moment-to-moment shifts in the plot.

We next took a closer look at the interplay between specific semantic structure components of the movie and particular brain states. As an illustrative example (Fig. 3), we display semantic context No. 152 that is associated with state 1 of the DN&AM model ($r = 0.19$). By bringing in contact meaning facets and movie-dependent brain responses, our analyses revealed that the concomitant part of the story was about Jenny and Forrest's wedding in front of their house (Fig. 3a, f). Further, although this semantic context was modeled to be distinct from semantic contexts at play at the beginning of the movie, these themes were inherently consistent, as they can be viewed as centering on "family bonds". By inspecting the semantic contexts' correlation with the 52 annotations, we confirmed that the scene took place at Gump's property ($r = 0.27$), while the most flagged emotion was happiness ($r = 0.17$; Fig. 3b). The external descriptions thus enriched the continuous narrative by consensus rater voting that indicated the presence of various features in the movie. Our results indicate that the links between the brain states and contextual movie information were well explained by external descriptions, and that these annotation-brain links were tied across extended time periods.

Based on the obtained semantic dissections, we used the contextual information of the story clips to explain brain activity from brain states capturing DN&AM patterns. The DN subregion contributions in brain state 1 (i.e., $\mu$ parameter estimates of the formed HMM) showed strong lateralization (Fig. 3c), while the precuneus/PCC made no relevant contribution in the left hemisphere but strong contributions in the right hemisphere. The most prominent role was played by the middle temporal gyrus in the right hemisphere. As to the subregion contributions of the AM (Fig. 3e), the right hemispheric anterior-amygdaloid-area subregion played the most prominent role. These

subregion signatures were externally linked to the positive emotions from the movie annotations. Overall, the DN and subcortical region reveals group-wise lateralization effects (Supplementary Figs. 15 and 16). Moreover, the estimated covariance relationships among subregions exposed the functional interplay between the DN and AM subregions (Fig. 3d). Within the DN, we observed that the functional coupling links among subregions of the right hemisphere were stronger than in the left hemisphere. Additionally, the left and right AM were coupled with the DN in different ways. Specifically, the AM included a group of subregions for which the coupling link directions were opposite in the left and right hemispheres. Yet, anatomically adjacent DN subregions showed neural activity effects in the same direction. Therefore, when processing positive emotions, the coupling links both between DN and AM subregions, and inside DN subregions exhibited a notable extent of lateralization effects. Taken together, these findings revealed that the links between the brain and movie contextual information were explained by specific subcortical subregion contributions and functional couplings.

### The deepest layers of the neural processing hierarchy preferentially track semantic movie contexts

We next turned from the subject level to the group level. By comparing the strength of each HMM instance's semantics-brain link, we corroborated the unique role of the DN in capturing semantic movie contexts (Fig. 4). Specifically, we trained a total of 210 HMM solutions (i.e., 15 subjects x 14 region-network combinations). The median correlation of the DN was the highest ($r = 0.163$), compared to the other six canonical networks under study (Cont: 0.144; DorsAttn: 0.141; Limbic: 0.133; SalVentAttn: 0.145; SomMot: 0.140; Vis: 0.126). Further, applying pairwise two-sample $t$ tests, all six pairs of group differences between the DN and other networks achieved statistical significance at $p < 0.0001$. After performing empirical permutation tests for each of 14 region-network combinations, all link strength values were larger than null model. This conclusion was also robust regardless of the choice of limbic region in the model and across different confirmatory analyses, which were performed with different text extraction methods, different numbers of semantic contexts, and different sources of movie text information (Supplementary Fig. 7). Considering the DN's stronger links with contextual information at the group-level, it may play an important role with regards to semantic association during naturalistic movie watching.

### Temporal dynamics reflect variance across subjects but consistency across neural processing layers

We next measured brain state dynamics at the group level based on the duration a given functional coupling state remains online. The average dwell time for each separate model describes the temporal switching properties or pattern volatility for each particular subject and for each particular region-network combination. i) Across subjects (Fig. 5a), we observed that the median dwell time for different region-network combinations ranged from 2.28 s (subject 05) to 25.31 s (subject 09). This divergence of dwell times might reflect idiosyncrasies in cognitive styles to approach movie content processing across different subjects. Additionally, we quantified the variance of dwell time profiles for each subject. The lowest standard deviation was 0.02 s (subject 05), and the largest one was 13.24 s (subject 15). ii) Across different region-network combinations (Fig. 5b), we again found that DN-tuned models showed the largest median durations of being continuously online (DN&AM: $t = 16.62$ s, DN&HC: $t = 16.87$ s). A two-sample t-test revealed that the choice of limbic regions did not change the average dwell time significantly ($p > 0.05$). The dwell times were most stable across different subjects in the Vis-tuned models (Vis&AM: $\sigma = 3.54$ s, Vis&HC: $\sigma = 3.59$ s). The full table of median and standard deviation values is shown in Supplementary Table 1. Significant differences were found between the average dwell times of DN-tuned models and those of Vis-

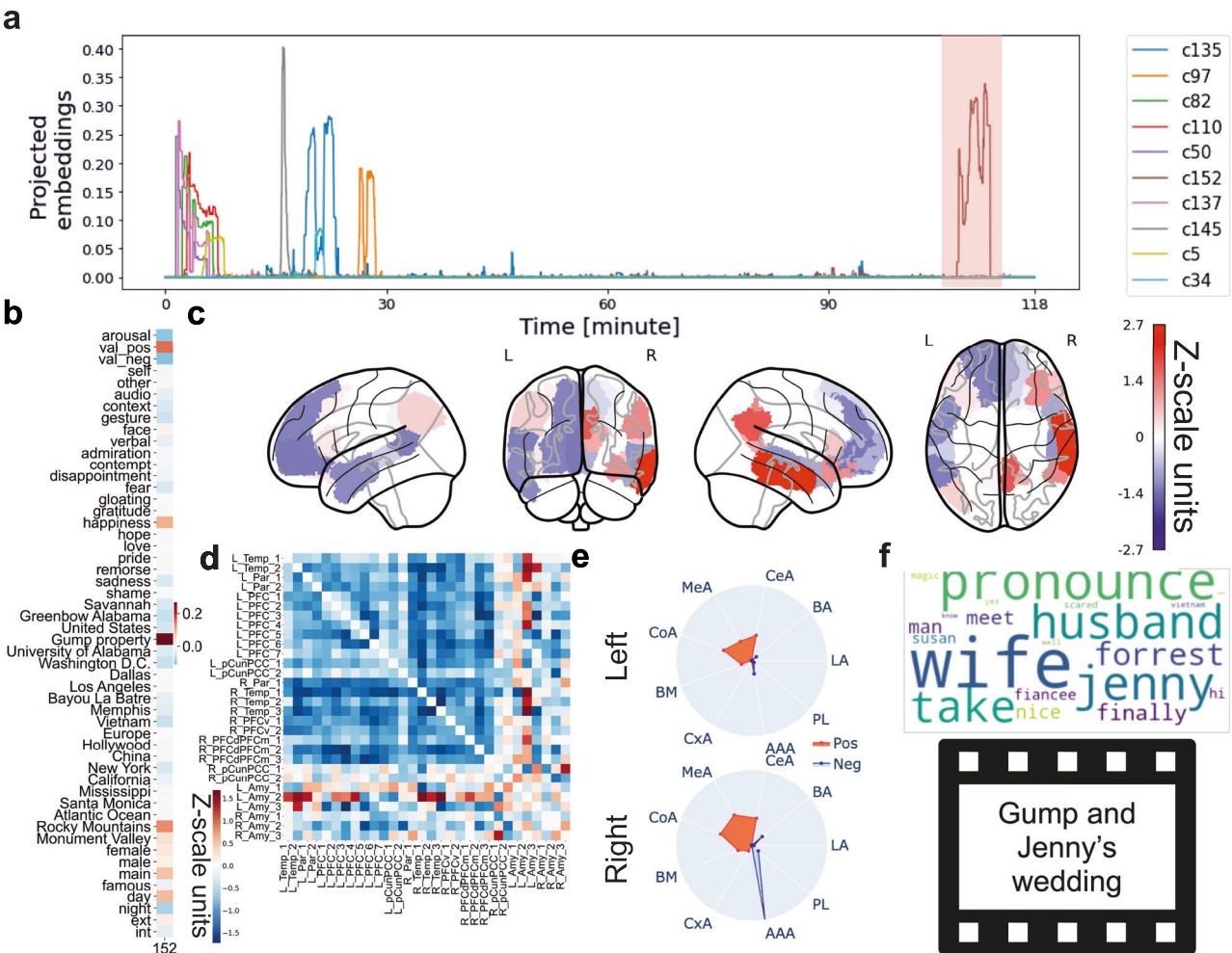

**Fig. 3 | Raters' and data-driven descriptions of story events show complementary links with brain state dynamics across 2-h movie.** We supplemented our semantics-brain links (i.e., bottom-up approach) with external-rater-curated annotations based on traditional neurocognitive concepts (i.e., top-down approach). **a** The 10 most correlated semantic contexts with brain state 1 of subject 1's DN&AM model (HMM). We highlighted the projected embeddings of our exemplary semantic context (No. 152) in pink. **b** Pearson's correlation between hand-made annotations and the semantic context No. 152. "Happiness", "positive valence", and "Gump property" were linked with the context features (for the other 199 semantic contexts, see Supplementary Fig. 2). **c** Brain renderings show DN&AM region-network contributions from dynamic brain state 1 (for the other states, see Supplementary Fig. 3). The most prominent subregion was the right middle temporal gyrus in the right hemisphere. **d** Functional coupling links among DN and AM subregions of state 1. The DN's subregion names are from Schaefer-Yeo 100 atlas (Temp: temporal; Par: parietal; PFC: prefrontal cortex; pCunPCC: precuneus

posterior cingulate cortex; PFCv/d/m: ventral/dorsal/medial prefrontal cortex). The bottom/rightmost six rows are principal components for the amygdala's left and right hemispheres. The left and right hemispheres (denoted as L and R in short names) of the AM related to DN in different patterns (for the other brain states, see Supplementary Fig. 4). **e** Model contributions of AM subregions differ in the left and right hemispheres. The radius value ranges from 0 to 1.6. Based on the radar plot, the right anterior-amygdaloid-area (AAA) was the most prominent subregion (for rest states, see Supplementary Fig. 5). **f** Contextual features of semantic context No. 152. Top: the wordcloud map. Word size indicates importance. The keywords include husband and wife. Bottom: the snapshot of a related part in the movie. LA Lateral-nucleus, BA Basal-nucleus, CeA Central-nucleus, MeA Medial-nucleus, CoA Cortical-nucleus, BM Accessory-Basal-nucleus, CxA Corticoamygdaloid-transition, AAA Anterior-amygdaloid-area, PL Paralaminar-nucleus. Source data are provided as a Source Data file.

tuned models ($p < 0.01$). These findings suggest that neural responses as modeled based on different region-network combinations largely depended on subject-specific dynamics. As a general trend, for lower-level networks, the timescales were less volatile across subjects. Importantly, the highly associative DN showed the longest dwell times across subjects and across brain states.

**DN showed different subregion signatures when paired with HC and AM**

We then characterized the across-subject commonalities of neural activity in the DN and its limbic partners observed during moment-to-moment changes in the movie at the group level. This was achieved by partitioning the subjects' brain activity timeseries into segments that belong to each of the four previously identified HMM states. Then, for

each of the four dynamic brain states, we concatenated the state-specific brain activity segments across subjects, which yielded four separate group-level neural activity time courses. We then applied partial least square regression (PLS-R) to extract the dominant signature that tracks how the 200 semantic context expressions and 52 external descriptions (i.e., 252 total input variables) explain subregion-level neural responses in the DN and its limbic co-activation partners (output variables). Throughout the cross validation tests (cf. methods; Supplementary Fig. 8), different states of brain embeddings achieved an average Pearson's correlation of 0.374 with the external descriptions embeddings. Following this approach, we directly associated, for each identified brain state (cf. above), the neural activity responses of the HC&DN subregions with hundreds of external descriptions at the group level.

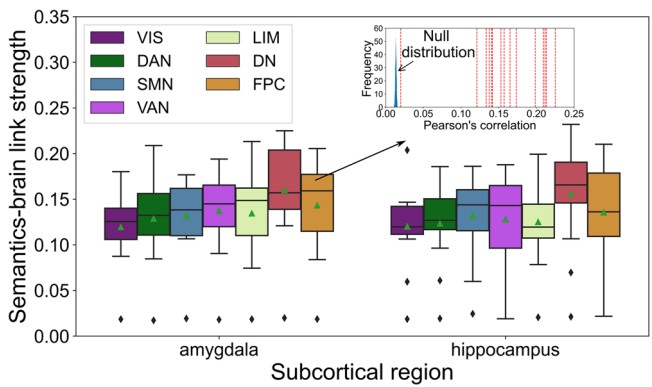

**Fig. 4 | Functional coupling dynamics in the deepest brain network layers track the movie narrative.** To compare the strength of semantics-brain links from lower (visual network, Vis) to higher network (default network, DN) layers in the brain, we computed the average Pearson's correlation strength between the presence of semantic context expressions and each of 15 subjects. Each color denotes one of the seven canonical functional networks (according to Schaefer-Yeo atlas definition, cf. methods). The collective amygdala (left) or hippocampus (right) subregions of the sub-neocortical system were jointly analyzed with these functional networks (cf. methods). The DN showed the most prominent median value, relative to the six other brain networks, which indicated this highly associative neural processing layer as most dominant in consistently tracking the semantic richness in the evolving movie narrative. Top right: an exemplary permutation test of the model. The red dash lines show the mean semantics-brain link strengths of 15 subjects, and the blue bars shows the null distribution by shuffling the state presence timeseries 1000 times (the DN&AM and DN&HC models were additionally depicted in Supplementary Fig. 11A, B, respectively). Boxplot: upper (lower) edge of the box is 25th (75th) percentile (interquartile distance); the middle line is the median value; the green triangle shows the mean value; the whiskers summarize the tail data points of the distribution of median semantics-brain associations. Short names for Schaefer-Yeo networks: VIS Visual network, SMN somatomotor network, DAN dorsal attention network, DN default network, LIM limbic network, VAN salience and ventral attention network, FPC Frontoparietal network. Source data are provided as a Source Data file.

Using the derived PLS-R solutions (again, one for each HMM brain state), we explored functional coupling signatures obtained separately for each of the two limbic partners (DN&HC: Fig. 6; DN&AM: Fig. 7). In the DN&HC model, we observed consistently large contributions of the PFC in the right hemisphere (Fig. 6b). Specifically, for states 1 and 3, the most prominent subregion was the ventral lateral PFC; for state 2, it was the dorsal lateral part of the PFC (dlPFC); for state 4, it was the dlPFC and medial PFC. The contributions of the left hemispheric subregions were relatively smaller and more uniform. In the DN&AM model, subregions in the precuneus and posterior cingulate cortex (PCC) showed the largest and most consistent contributions (Fig. 7b). For states 1 and 4, their direction was positive. For state 2, the direction was negative, and the ventral part of precuneus/PCC had larger contributions. In contrast, for state 3, the dorsal part showed large and negative contributions. Concluding on an overarching trend, in naturalistic movie watching, the PFC appeared to play an active role together with the HC, while the precuneus/PCC were more active in conjunction with the AM.

To further detail the DN signatures across the four brain states, we next attended to the coupling relationships with their limbic partner regions. For the DN&HC model (Fig. 6a), in states 1 and 2, the CA1 and CA3 played prominent roles. Additionally, we observed that the contributions of the anterior subregions of the HC mostly dwarfted those of its posterior parts. States 1 and 2 were mostly linked with places in external descriptions (Fig. 6d). For state 3, the prominent subregions included subiculum head, presubiculum head, and parasubiculum. For state 4, the most important subregions were CA1-4, granule cell layer of the dentate gyrus, molecular layer, parasubiculum, fimbriae, and HC tail. The tapped semantic contexts (Fig. 6c) mainly reflect events involving different movie characters. The top semantic contexts across four signatures covered all main characters, including Forrest Gump, Lieutenant Dan, Gump's Mom, and Jenny. We also automatically isolated music contexts by the keyword "gesang" (bottom row, meaning singing in German). Further, in states 3 and 4, we flagged two categories of annotations describing the surrounding environment in the

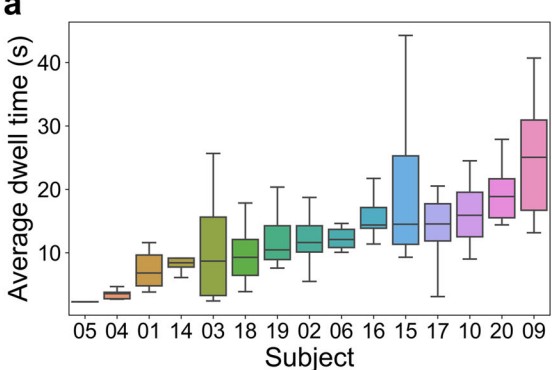

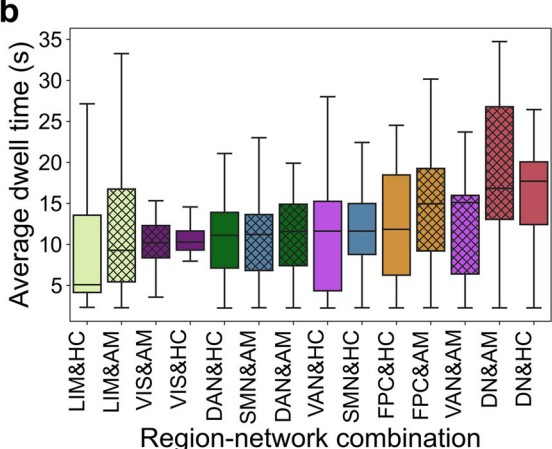

**Fig. 5 | Temporal profiles of dynamic brain states are characteristic for individual subjects and region-network combinations.** To carve out the time scales of how distinct brain states click in and detach during movie watching, we quantified the dwell times of the states from each hidden Markov model (HMM) solution: we computed the average time of occupancy, between locking in and abandoning a given state, across the four states within an HMM solution at hand. We examined the 210 total trained HMM models in two ways: **a** Across subjects, the median value (the center bar in each box) ranges from 2.28 s to 25.31 s (x-axis was ordered from lowest to highest based on the median value, the same for (**b**)). The 15 subjects (assigned with random color) demonstrated distinct neural processing mechanisms tracking movie content. **b** Across region-network combinations (amygdala[AM]-network models in hash marked, hippocampus[HC]-network

models in plane, the color is according to Schaefer-Yeo atlas[38]), DN-tuned models with AM or HC coactivation partners showed the longest dwell time, while analogous VIS-tuned and LIM-tuned models showed the most consistently low dwell times with the smallest standard deviation. Again, the collective results witness the higher-order integration function of the DN to play a dominant role in movie watching. Boxplot: upper (lower) edge of the box is 25th (75th) percentile (interquartile distance); the middle line is the median value; the whiskers summarize the extreme data points of the distribution of median semantics-brain associations. Short names for Schaefer-Yeo networks: VIS Visual network, SMN somatomotor network, DAN dorsal attention network, DN default network, LIM limbic network, VAN salience and ventral attention network, FPC Frontoparietal network. Source data are provided as a Source Data file.

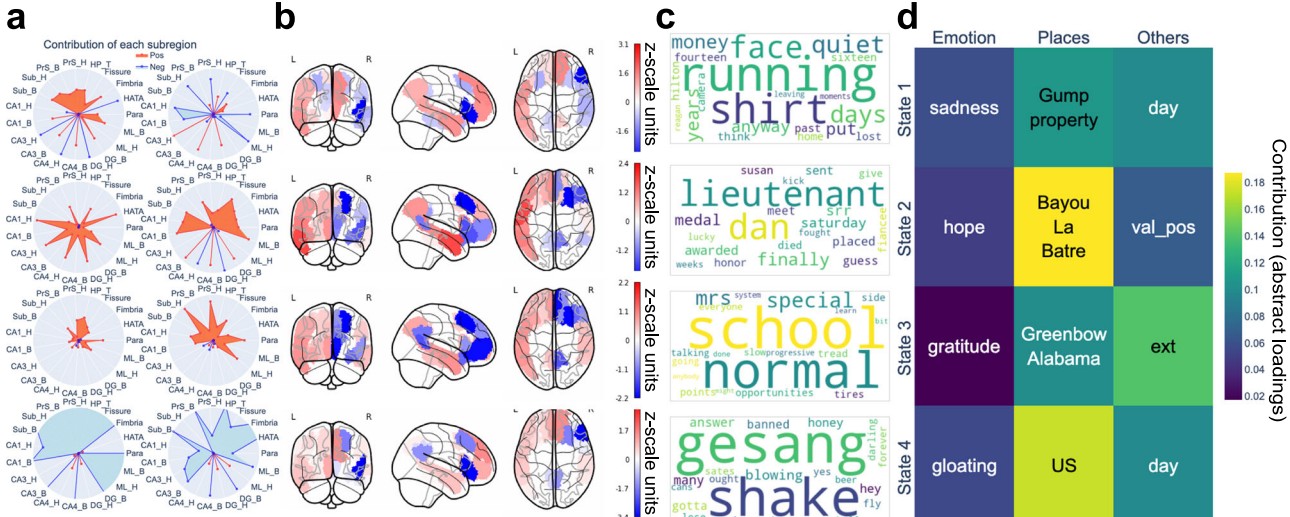

**Fig. 6 | Default network (DN) and hippocampus (HC) coupling signatures are linked to narrative events and external descriptions.** Using partial least squares regression (PLS-R) modeling, we identified coherent cross-associations of neural responses with both data-led semantic contexts and hand-selected annotations. Each row presents results for one derived PLS-R solution corresponding to one of the four brain states (cf. methods). **a** Contributions of the 19 left and 19 right subregions of the HC. The radius ranges from 0 (center) to 1.6 (outer circle). Generally, left and right hemispheres play different roles in the neural signatures. **b** The DN subregion contributions, covarying with the HC, highlight the prefrontal cortex (PFC). **c** The semantic context elements that are best explained by each DN&HC coupling signature. The underlying contextual events were related to various movie characters: 1. A man who made money from T-shirts that Gump used when running; 2. Lieutenant Dan; 3. Gump's Mom; 4. Jenny sang songs nakedly. **d** Among the 52 external descriptions, the three strongest annotation entries related to the three categories: "emotion", "place", and "others". The contributions of places were the largest. H head, B body, T tail, CA dentate gyrus and Cornu Ammonis, Para para-subiculum, HATA hippocampal amygdala transition area, HP hippocampus, PrS presubiculum, Sub subiculum, ML molecular layer; DG granule cell layer of the dentate gyrus (GC-DG-ML). Source data are provided as a Source Data file.

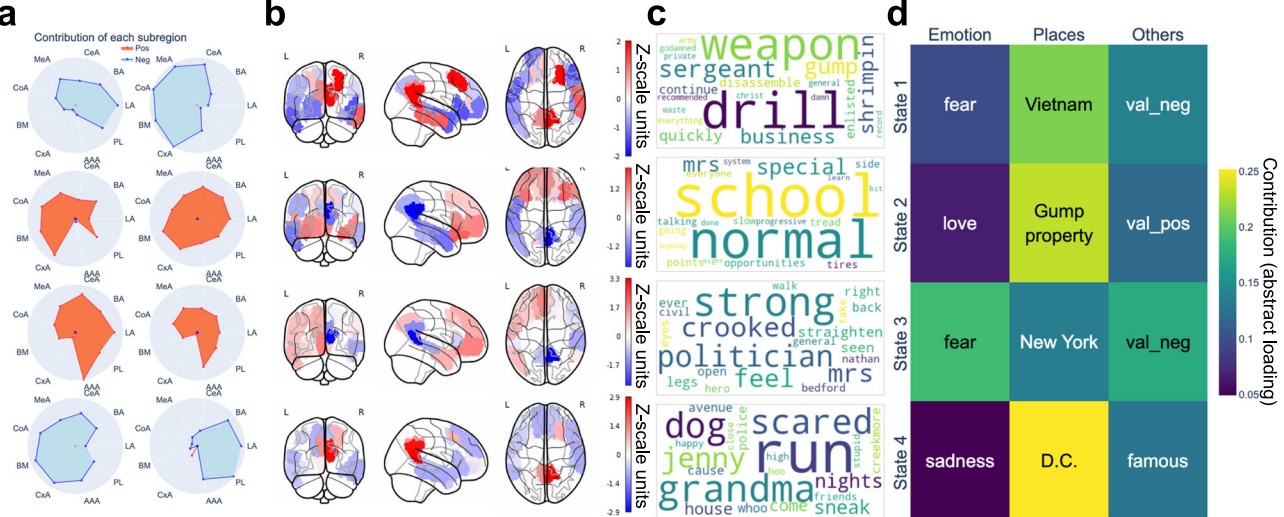

**Fig. 7 | Default network (DN) and amygdala (AM) coupling signatures are linked to narrative events and external descriptions.** Using partial least squares regression (PLS-R) modeling, we identified coherent cross-associations of neural responses with both data-led semantic contexts and hand-selected annotations. Each row presents results for one PLS-R solution corresponding to one of the four brain states (cf. methods). **a** Contributions of the 9 left and 9 right subregions of the AM. The radius ranges from 0 (center) to 1.6 (outer circle). Generally, left and right hemispheres play different roles in the neural signatures. **b** The DN subregion contributions, covarying with AM. Precuneus and posterior cingulate cortex (PCC) played an important role. **c** The semantic context elements that are best explained by each DN&AM coupling signature. Contexts that were related to emotions were associated with AM: 1. Vietnam war; 2. Gump's mom; 3 Gump was treated; 4. Gump ran away from a dog. **d** The emotional entry "fear" appeared to be larger with AM. AM-tuned DN activity focused more on emotions, especially "fear". LA Lateral-nucleus, BA Basal-nucleus, CeA Central-nucleus, MeA Medial-nucleus, CoA Cortical-nucleus, BM Accessory-Basal-nucleus, CxA Corticoamygdaloid-transition, AAA Anterior-amygdaloid-area, PL Paralaminar-nucleus. Source data are provided as a Source Data file.

movie: the placement of the camera (exterior/interior, cf. methods) and the time of the day (day/night). These findings suggest that, functionally interlocked with the highly associative DN, HC responses were mainly associated with places, characters, and environmental features during uninterrupted naturalistic stimulation.

For the DN&AM model (Fig. 7a), the AM subregion signatures corresponding to brain states 1 and 4 showed concurrent effects in the left and right brain hemispheres. In state 2, several cortical-nucleus-related subregions contributed most and bilaterally; and in state 3, it was the anterior-amygdaloid-area subregion bilaterally. Regarding

associations with external descriptions, fear was highlighted in states 1 and 3, this emotion tag was the strongest emotional description in state 3 (Fig. 7d). The emotional concepts flagged for states 2 and 4 instead were love and sadness. Complementing the single word descriptions of classically studied emotions, our derived semantic contexts also reflected emotion-related story clips (Fig. 7c), including war scenes, family bonds, hospital treatment, and dog-chasing scenes. Taken together, functionally liaised with the highly associative DN, we linked several classes of emotional processes to specific AM subregion signatures, which also tracked emotionally-laden scenes along the movie story.

## Discussion

In our everyday lives, conscious awareness and episodic narratives animate important classes of human cognition[41]. To step closer to uncovering the neurocognitive basis of our daily experiences[3], naturalistic neuroscience using movie watching has started to reveal untapped aspects of brain processes in a more ecologically valid outlet, compared to classical experimental paradigms[6,42,43]. For the purpose of the present investigation, we propose an analytical framework that translates natural language processing algorithms from machine learning (cf.[29]) to isolate and integrate the constituent structural elements of the semantics that scaffold the movie. The derived semantic descriptions of the narrative then directly informed fine mapping of the neural responses to the unfolding story line. Capitalizing on 2-h length brain-scanning during which participants watched the movie Forrest Gump (>50,000 individual maps of whole-brain activity responses), we were able to quantify moment-to-moment shifts in brain coupling dynamics by a head-to-head comparison of seven major canonical networks at a single-subject level. Through the careful exploration of >200,000 HMM solutions, we here demonstrate that engagement of the highly associative DN, rather than the six other probed neocortical networks, was intimately coupled with events of the evolving movie narrative.

Semantic processing has been proposed to be essential for human higher-order functions[44]. Due to the brain's specific energy and wiring constraints[45], efficient high-level information integration and retrieval functions are needed to realize many types of advanced neural processes. To clarify the definition of semantics, the word "semantic" has been used incongruently in different fields, including semantic memory, semantic processing, and linguistics[46–50]. Here, the use of the term is closely linked to the notion of "structured knowledge of the external world", as proposed by Binder and colleagues[47]. We thus executed a survey of seven functional brain networks regarding the strengths of their semantic context-brain links. In so doing, we established that the DN's neural responses were most intimately linked with momentary shifts in the semantic contexts throughout the whole movie. In addition, DN activity has previously been reported to reflect the people's own interpretation of narratives[51,52] and to link with social perception[53]. Based on this previous progress, we argue here that it is important to obtain single subject-level resolution on such brain dynamics and offer a dedicated analytical toolkit and interpretation framework.

Our results also extended descriptions of the chronoarchitecture that undergirds the neural processing dynamics in previous naturalistic neuroscience studies[54,55]. The timescales describing how often and how long a dynamic brain state flips online have been termed "temporal receptive windows" (TRW[8]). Early hints suggested that TRWs may differ across separate circuit layers of the neural processing hierarchy[8,56,57]. In our systematic assay of brain network involvements during movie appraisal, spanning the neural processing hierarchy across seven spatiotemporally coherent functional systems[36], the DN exhibited the longest-lasting TRWs. In contrast, lower sensory networks typically showed shorter TRWs. The observed divergences in TRWs between the potentially deepest (highly associative DN) and some of the shallowest (early sensory) network layers were believed to

be a precondition of successful deep processing of complex semantic information whose presentation itself often takes a few seconds in a movie[5,54]. Long and rich sequences of nested events occurring in movie plots were also shown in a previous fMRI experiment to trigger reliable responses in the DN, especially its parts of the PMC, rather than any parts of lower-level network systems[8]. The observed TRW of the DN in this earlier study was comparable to the DN's TRW that we observed here (~12 s on average). In sum, our derived temporal and spatial configurations of functional couplings hardened and added nuances to previous hints at the DN in evaluation of environmental semantics.

After showing in our analyses that the DN emerged to be intimately related to tracking semantic structure in real-world stories, we aimed to detail the DN's neural activity responses as they liaise with key partners from the limbic system[58]. Our study uncovered and quantitatively characterized two core mechanisms of collaboration with sub-neocortical brain systems: one DN pattern highlighting the partnership with dedicated AM subregions and another one active in concert with dedicated HC subregions (Figs. 6, 7). For each functional pattern of dynamic couplings, we modeled four separate neural signatures with their flanking semantic descriptions to be able to capture complementary aspects of network dynamics sensitive to the story. We found that the DN subregions and semantic label contributions have significant differences when paired with different subcortical partners (Supplementary Fig. 14). In the following paragraphs, we will delve into the subtle functional and anatomical differences between the DN and amygdala as well as the DN and hippocampus.

We found that the functional coupling interactions between DN and AM (DN&AM signatures) were associated with emotional annotations and key semantic elements of the movie narrative. In particular, designated parts of the DN's PMC contributed disproportionately to the overall constellation of neural activity responses to the vivid depiction of movie scenes. Previous brain-imaging studies claimed that PMC activity changes track the fantasy content in movies such as Alice in Wonderland[59,60] as well as the surprising content such as in sports matches or TV shows[48,61]. Our analyses invigorated a central role for the PMC in processing naturalistic narratives. We laid out these functional implications across several unique signatures of DN&AM interaction, which emerged especially in the context of emotional movie events. We argue that previous reports on such PMC implications were limited by linking movie content to mostly a single cognitive concept (e.g., fantasy or surprise). However, such earlier reports may be re-interpreted as pointing to a broader role of the DN in pooling, elaborating, and binding brain-wide information[62,63]. Collectively, to reconcile present and previous findings, the DN makes critical contributions to tracking and integrating amygdala-preprocessed, emotionally evocative turning points of the movie narrative. Especially for real-world-like naturalistic stimuli, this mechanism of adjusting to emotionally amplified information in the environment may have evolved in humans for coping with upcoming challenges and changes in the external world[64].

Further, our results of cortical functional interplay with limbic partners speak to why and how emotional semantics were tracked by our detected DN&AM signatures. First, the AM has long been treated as the heart of emotion processing in the brain[65,66]. Extending such earlier findings to subregion granularity, we now brought to the surface the complementary lateralization effects from AM activity. The subregions with stronger contributions in the left-hemispheric amygdala usually showed weaker functional contributions in the right hemispheric amygdala. Conversely, the amygdala subregions with weaker contributions in the left hemisphere tended to play stronger roles in their counterparts in the right hemisphere. Similar asymmetric effects of neural responses were also discovered in a previous emotion and word linkage experiment from electrophysiological recordings in humans[67]. Therefore, our lateralization findings in the AM further confirmed and explained how emotionally

evocative semantic contexts are tied to the subregion-specific lateralization effects in the AM.

Second, in line with recent reasonings[64,68], our discovered movie-induced coupling interactions between the PMC and specific AM subregions dovetail with their putative implications in external environment monitoring, especially significance detection and self-relevance evaluation. Moreover, the derived external descriptions offered rich contextualization for neuroscientific interpretation of the extracted limbic–neocortical interaction patterns that appear to directly speak to the attention deployment theory of emotion control[69–71]. That is, our findings may reflect attention reallocation mechanisms that came to bear when subjects were viewing unpleasant movie scenes[43]. For example, "valence" became apparent as one of the leading annotations in three out of four DN&AM signatures: Indeed, the functional coupling signatures associated with negative valence annotations related to more unpleasant semantic scenes of the movie plot (e.g., war). In the face of complex affective semantics in a real-world simulation experiment, we thus linked adaptive emotion regulation with flanking functional coupling changes between the highly associative PMC and dedicated AM circuits.

More broadly, our collective findings motivate an extension to the traditional AM survival theory by means of higher semantic reflection. According to the classical view[72], humans show intuitive responses to sudden changes in the ambient environment. For example, if a person saw a bear chasing, their adrenaline level would surge automatically, as an instance of a fight-or-flight reaction mediated by the sympathetic nervous system. Revising this classic notion, the AM survival theory may benefit from integration with neural processes subserved by the recently evolved deepest neural processing layers: based on continuous conscious awareness of consistency or discrepancies of environmental features, the DN may potentially liaise with dedicated AM subregions to calibrate the scanning of the external world for self-relevant information and otherwise emotionally evocative cues. In this way, emotionally edited sensory information can be instrumental to the higher association circuits by giving color to a vast number of candidate semantic interpretations and by effectively directing the allocation of attentional resources based on an evaluation system of significance for the organism[73–75]. After detecting behaviorally relevant information in the environment, the human brain also needs to integrate semantic knowledge into the memory system to store information and compare it against past experiences to help with upcoming decisions on how to act on the world.

Across the delineated DN&HC signatures, medial and lateral parts of the PFC showed functional coupling with designated HC subregions, as another core limbic partner of the higher association cortex. HC-PFC pathways have been discussed before to be involved in episodic scene construction and memory[76]. According to previous reflections, the PFC is implicated in the suppression of content-independent stimuli to boost information retrieval from the environment. Instead, the HC probably subserves retrieval and organization of content-related memory[76]. Our HC subregion-level delineation showed that neural responses of CA1-4 and subiculum (especially the head segments) were functionally interlocked with PFC activity responses. This observation confirms and details previous reports on the HC-PFC pathway: it is well-established that the PFC receives direct axonal projections from the hippocampal subiculum and CA1 in both animals[77–79] and humans[80,81]. In these studies, the medial PFC, as opposed to its lateral parts, was typically more emphasized for its dense fiber bundle connections to the HC subregions that our quantitative analyses here spotlight during movie engagement[82].

In our analyses, subiculum activity, among the HC subregions, was prominent in the signature associated with the type of locations in which movie scenes took place. Functional engagements of the CA1 subregions, in turn, dominated the signatures that were linked with the time of the day of movie scenes. The subiculum is believed to assist in identifying and binding spatial boundaries in the environment according to previous hypotheses and experiments in animals[83,84]. The annotations pertaining to open-space vs closed-space of movie events can be viewed as a defining feature of the encountered spatial scenes. Indeed, neurons in the CA1 have been reported to be sensitive to bright vs dark environments during experiments on light condition transition in rodents[85]. The same sensitivity to ambient light conditions was also revealed by our time of the day annotation. Therefore, by linking it with previous invasive single cell recordings in animals, our functional analysis in humans extended abstract aspects of environmental information processing mechanism (boundary and light conditions) to the naturalistic setting. In sum, our analyses disclosed DN&HC coupling constellations that may assist more abstract forms of information processing in the naturalistic setting.

More generally, our findings provide valuable clues to explaining the higher-order functional capacities that underlie hippocampally-assisted semantic reasoning. The stronger functional contributions for the anterior parts of the HC may relate to previous results on the anterior HC subregions' involvement in semantic memory processing over and above its posterior subregions[86–88]. In recent reasoning around functional roles of semantic memory, Strange et al.[89] suggested that it provides the flexibility needed for conscious planning by encoding abstract features and forming higher-order memory representation. Previous experiments on semantic memory in humans confirmed that HC activity increases during transitions between movie events in both traditional experiment events[90,91] and naturalistic movie stimuli[54,92,93]. Here, by quantitatively dissecting the meaningful elements of the story, our findings complement the temporal alignment between HC activity changes and movie event offsets.

It is important to acknowledge some limitations when interpreting our results and conclusions. First, appreciation of findings from the studyforrest dataset need to take into account the fact that the majority of the subjects had previously watched the movie and are not English native speakers. This circumstance could potentially color some of the obtained results. For example, the familiarity of the subjects with the movie could lead to a diminished surprise level. Second, another limitation of the studyforrest dataset is that it does not allow for cues on individual-level semantic processing itself, which is however a crucial aspect in understanding and interpreting language. The lack of individual feedback restricts our ability to explore further the subjects' response to the movie. This shortcoming could be addressed by extending the studyforrest repository to include the participant's own interpretation of the events in the movie material in a timepoint-by-timepoint fashion. Indeed, Saalasti et al.[94] recorded subjects' thoughts as a reflection of "what comes into your mind." Additionally, Baldassano et al.[54] asked subjects to recall the movie content to assess "what do you remember".

Painting a broader canvas to summarize, our analytical framework opened a window to identify two distinct mechanisms of how the DN dynamically partners with microanatomical subregions of the AM and HC to trace semantic salience and their changes in the environment, by sifting through a compilation of >20,000 HMM estimations across seven large-scale networks. In this way, we offer explanations of how some of the deepest brain network layers of the human brain support the active search for meaning and valuable information in the external world—a precondition for judicious choice of candidate actions from the behavioral repertoire[25,64].

## Methods

### Rationale and workflow summary

Previous naturalistic stimulation studies have largely focused on modeling the neural responses which are elicited by a movie. In such earlier approaches, the fine-grained information contained in the movie material itself was typically neglected or not fully analyzed. Directly capitalizing on the continuous visual-auditory stimulation

may be better achieved by bringing to bear untapped analysis frameworks that pool across distributed brain activity responses and the movie events which invoke it, by invoking comprehensive descriptions of salient movie events. To achieve this goal, we proposed two key steps. First, we brought to bear state-of-the-art ML tools in their home territory, that is, multimedia information analysis. Second, the contribution of limbic areas was rarely linked with neocortical activities in explicit quantitative modeling on naturalistic stimulation, despite recent progress in anatomy[37]. As pointed out before[95], the cortico-centric interpretational focus (which treated the cortical activity as an isolated process) of brain-imaging discoveries might not explain the full picture of the brain dynamics. To delineate and annotate the functional coupling dynamics of both subcortical-limbic and neocortical circuits, our elected analytical framework extracted embedding representations of the movie plot itself.

In recent years, the conceptualization of the subcortical limbic system as a component closer to the higher association cortex than previously thought has led to increased interest in studying the relationships between the cortex and the limbic key nodes amygdala and hippocampus. These sub-neocortical regions, through their coordination, play a crucial role in the elaboration of emotions, memories, and stimulus-value associations. The anatomy of the default network closely resembles the unitary model of the limbic system[37], making it an essential component to consider when studying the underlying mechanisms of emotion, memory, and behavior during naturalistic movie appraisal in humans. The interactions among these regions should shed new light on the mechanisms underlying competition for limited computational resources and how the brain captures distinct elements of semantic information.

Specifically, we ported NLP techniques from machine learning to imaging neuroscience for explicit movie narrative modeling. Put differently, we aimed to leverage the most immediately human interpretable feature of the rich multimedia movie data—the language information. To carefully track the evolving movie plot, the trends of word usage in movie subtitles and verbalized descriptions provided the basis for enrichment of the concurrent neural activity responses. We extracted a rich portfolio of 200 unique semantic contexts whose occurrence across the 2-h length movie Forrest Gump served as a proxy for movie events. The unusual wealth of time series data offered by the studyforrest dataset (54,145 total timepoints, combined from 15 subjects) enabled us to quantitatively profile nuanced region-network functional coupling dynamics. Explicit modeling of network dynamics, delineated via our HMMs, was performed at the single-subject level (where each subject's brain recordings were analyzed separately) and specific to a particular canonical network (where neural signatures of subregions in each canonical network were examined in an individual model). After isolating functional coupling regimes of canonical networks and limbic subregions, we were able to characterize the subregions' neurocognitive roles by association with 200 semantic contexts and 52 human-curated annotations, spanning from properties of movie scenes (e.g., places, time of the day) to descriptions of movie characters (e.g., character identity, their emotion expressions, the valence of these). Across conducted analyses, we charted >20,000 HMM solutions under the movie appraisal. With this extended space of HMM solutions, we delineated subregion-subregion interactions of a total of 14 limbic-neocortex views, including their functional coupling patterns, specific subregion contributions, and timepoint-specific presence across the evolving movie narrative.

## Data resources

**Functional brain images.** 15 subjects were recruited (mean age: 29.4 years of age, ranging from 21 to 39, 6 females) for both i) audio-visual and ii) audio-only movie-watching during fMRI scanning as part of the studyforrest protocol. All participants were right-handed German native speakers. This study was performed under the Ethical approval

from the Ethics Committee of the Otto-von-Guericke University, Germany, and the informed consent of all participants.

In the audio-visual naturalistic stimulation of the studyforrest project[96], the movie "Forrest Gump"[97] was segmented into eight cuts (~15 min) using the same method that was previously applied to the audio-only version of the movie[98]. Accordingly, the identical approach as in the original study was adopted regarding the transition between segments and their synchronization with the acquisition signal of the fMRI scanner. The boundary between segments was determined to ensure that fMRI volume acquisition was aligned with the movie across all eight cuts (for details, see ref. 99; in particular, Table 1 and Fig. 3a). Briefly, except for the first cut, each of the eight segments began by fading in an excerpt of ~6 s (three fMRI volume acquisitions) from the end of the preceding segment. Similarly, except for the last cut, each segment ended by fading out an excerpt of ~10 s (five fMRI volume acquisitions) from the beginning of the following segment. The start of each segment was synchronized with the acquisition signal of the fMRI scanner. Here, to ensure the alignment in time between the fMRI activity time series and the movie, we discarded the fMRI timepoints corresponding to the repeating start and end excerpts for each individual cut.

In the experiment, all subjects watched the movie in the 3T Achieva scanner (Philips Medical Systems) with a 32-channel head coil. 14 subjects had previously watched the movie, and the additional subject had previously been exposed to the audio-only descriptions. The dataset also provides high-resolution T1-weighted structural images[98] that were acquired using a 3D turbo field echo sequence. The voxel size of the acquisition was 0.7 mm along with a 384 in-plane reconstruction matrix (0.67 mm isotropic resolution). The other parameters were as follows: TR = 2500 ms, TE = 5.7 ms, TI = 900 ms, flip angle = 8°, FOV = 191.8 × 256 × 256 mm, bandwidth = 144.4 Hz/px, sense reduction AP = 1.2, RL = 2.0. On the other hand, the functional images[96] were acquired with a gradient-echo, T2*-weighted EPI sequence (TR = 2000 ms, TE = 30 ms, flip angle = 90°, axial slices = 35, thickness = 3.0 mm, gap = 10%, FOV = 240 × 240 mm, voxel size = 3 33 mm). To put the topmost slice on the superior edge of the brain, the slices were ordered in AC−PC orientation with the SmartExam system (Philips).

## Human-curated annotations

To generate consensus descriptions of the emotions portrayed in the movie Forrest Gump[99], nine observers (all female) from the same student population have annotated the audio-visual version of the movie[100] with a total of 52 indicators. None of the raters took part in the previous brain-imaging study. The movie has been segmented into 205 cinematographic scenes, which were annotated by each observer in randomized order to mitigate carryover effects. For each instance of expressed emotion, observers indicated the start and end time in seconds, as well as the associated emotion. The independent raters also indicated the way in which the particular emotion is expressed, the values of three forced-choice variables: valence (positive vs negative), arousal (high vs low), and direction (whether the emotion is oriented towards oneself—character expressing it—or someone else). To-be-distinguished sources of the emotional cues included facial expressions, body language, abstract contextual information, verbal, and non-verbal audio cues. Offset cues included a change of emotion, a neutral state, the character leaving the scene, and the end of a scene. Valence, arousal, and direction were used to coarsely characterize the emotion in line with dimensional models of emotion[101].

In contrast to the indicators of arousal, valence (positive and negative) and direction (self and other) were deemed non-exclusive. That is, a movie character could be reported as portraying both positive and negative emotions simultaneously. Similarly, the acting character's emotion could be both self-directed and oriented towards others. As to discrete emotional labels, 22 categories were derived

from a model developed by Ortony, Clore[102], including anger/rage, fear, happiness, love, and sadness—the top set of emotions assigned in the majority of emotional events in the movie (for an exhaustive list, see Table 3 in Labs et al.[100]). The posthoc raters were instructed to assign an emotional tag to a movie event only if it perfectly matched one of the categories. An automated quality control procedure was used to check for errors and potential issues in the curated annotations.

The studyforrest dataset also provides annotation of the physical location in which each scene takes place[103], its type (interior or exterior), as well as the time of day (day or night). The movie scenes were annotated by two individuals, including a rather with an academic background in filmmaking. For the time of day annotation, a scene was labeled as "day" whenever it was illuminated by sunlight, including twilight, and labeled as "night" when there was no sunlight.

Inter-observer agreement (IOA) time series were then computed for each character and each emotional attribute. To this end, we indicated the proportion of total observers who identified the presence of a given attribute at each point in time. Therefore, IOA time series generally range from 0 to 1. However, IOA time series for the arousal, valence and direction indicators were obtained by subtracting from each other the IOA time series corresponding to the presence of both extremes, for example, IOA for low-arousal was subtracted from the IOA for high-arousal, resulting in an interval of −1 to +1, with two extremes indicating perfect agreement for low and high arousal, respectively. All IOA time series were then binarized using an absolute threshold of 0.5 to segment the movie into emotional episodes for each character. An emotional episode was defined as a period during which at least one emotional attribute exceeded the IOA threshold. Arousal and positive and negative valence values were attributed to each episode by computing their median IOA values across the length of the emotional episode. Here, to ensure a high level of consistency across observers in the annotations under consideration, we used these emotional episodes detected by the majority of observers to generate time series reflecting the presence of specific emotional expression features. For arousal and positive and negative valence, aggregate IOA time series were generated by assigning the maximal IOA value to each timepoint (2-s windows between the fMRI acquisitions) across all emotional episodes and characters. For all other emotional attributes, binary time series were produced to reflect their presence (0: absent, 1: present) at a given time step across all emotional episodes and characters. Analogously, binary time series were produced to reflect the presence (0: absent, 1: present) of each of the four location annotations (interior, exterior, day, and night) at a given time step in the movie.

## Audio text material from the movie

Two complementary sources of verbalized content provided the basis for our natural language processing approaches (cf. below) for integration with the functional coupling dynamics detected during movie watching. Notably, the volume of the original soundtrack is automatically scaled so that the narrations are easily perceptible. First, time-aligned movie subtitles in were also provided by the studyforrest resource. The subtitles from the original movie reflect the speech of the characters, as well as the narration of the movie plot by an off-screen voice from the main character Forrest Gump. Second, the analyzed material also includes the movie description[104] used by Hanke and colleagues[98] for audio-only experiments. This audio-only content is largely like the dubbed soundtrack of the movie. The audio description also includes interweaved narrations by a male speaker, mainly describing the visual components of the movie. These fill in verbal explanations in the movie scenes that do not otherwise contain any dialog, off-screen speech, or other related audio information, complementing the subtitles.

**Brain-imaging data preprocessing.** All of the template brain images and precomputed transformations contained in the studyforrest dataset (https://github.com/psychoinformatics-de/studyforrest-data-templatetransforms) were generated using the fMRIB Software Library FSL [105]; fsl.fmrib.ox.ac.uk. Regarding brain structure, non-brain tissue was removed from the T1-weighted structural image using the Brain Extraction Tool BET[106] with a robust iterative brain center estimation. Then, an affine transformation was computed each subject's native structural space to standard MNI space using the MNI152 template image with FMRIB's Linear Image Registration Tool FLIRT[107,108]. The inverse transformation matrix was also computed, which maps from standard MNI space to each subject's individual anatomical space. Finally, the subject's structural template was bias-field-corrected using FAST[109].

Regarding brain function, the BOLD images were motion-corrected using MCFLIRT[108] in two iterative stages using the mean BOLD volume, followed by the skull-stripped mean motion-corrected volume as reference. Each image was aligned separately in a single step using a combination of the two transformations. The aligned fMRI images, as well as the code used to obtain them have been made openly available (https://github.com/psychoinformatics-de/studyforrest-data-aligned). The same reference brain template was then used to compute a rigid-body transformation from each subject's structural space, using the bias field-corrected template, to its BOLD space, once again by means of FLIRT[107,108].

**Workflow for subregion neural signal extraction.** Using FLIRT and the available native-space-to-common-space transformation matrices (cf. above), we mapped the Schaefer 2018 parcellation (https://github.com/ThomasYeoLab/CBIG/tree/master/stable_projects/brain_parcellation/Schaefer2018_LocalGlobal/Parcellations[38]); to each subject's brain space, with a resolution of 100 regions of interest (ROI). To extract voxel averaged functional signals, ROIs served as topological masks, were used to average the signals from all the voxels that belonged to one spatial definition. As a result, we obtained as many functional BOLD signal variables as there are target subregions in the ROI set.

Regarding the limbic partner regions, we performed the segmentation of microanatomically defined HC and AM subregions based on T1 data in the studyforrest dataset via the segementHA function of free-surfer v7.1.1 (https://surfer.nmr.mgh.harvard.edu/fswiki/HippocampalSubfieldsAndNucleiOfAmygdala[110]). The exemplary mask figures are shown in Supplementary Fig. 9. In this way, we obtained the micro-anatomically defined labels for 38 hippocampal (19 per hemisphere) and 18 amygdalar subregions (9 per hemisphere) unique for each subject's brain anatomy. For the hippocampus subregions' segmentation, we adopted the head, body, and tail parcellation[111]. For the amygdala, we selected the nuclei labels proposed by Saygin and colleagues[112]. As a result, the functional signals of HC and AM for both the left and right hemispheres were extracted based on the ROIs defined by subject-specific brain anatomy.

We then detrended and z-scored the fMRI signal separately across timepoints belonging to each of the eight imaging segments (cf. above). All these data slicing and dicing steps were realized using functions from the nilearn package (https://nilearn.github.io/stable/index.html).

## Natural language pipeline to mine movie semantic dimensions

To enable quantitative modeling of underlying constituent movie events, we constructed a bag-of-words encoding of both the subtitles and the audio-only descriptions of the movie. The two kinds of text information naturally captured two complementary abstraction levels of the ongoing narrative: the movie subtitles cover the dialogs between characters, whereas context descriptions cover the detailed explanations of what is happening on the screen. To this end, we

initially removed all the punctuation marks and turned all letters into lower case. We next removed stop words, which is a typical workflow step for NLP preprocessing that aims to increase the performance of the downstream analysis methods. The stop words are a collection of commonly used words without much contribution to text understanding (e.g., prepositions, pronouns), which we drew from the Natural Language Toolkit Python package (NLTK; https://www.nltk.org/). We obtained a bag-of-words representation: a working dictionary of 1558 unique words for subtitles and another dictionary of 1503 words for descriptions. Pooling the resulting text information across the 3543 timepoints (indexing 2-s movie chunks), the word count matrix $M_{sub}$ for subtitles carried 3543 timepoints × 1558 vocabulary entries, and the word count matrix for description $M_{des}$ carried 3543 timepoints × 1503 vocabulary entries.

To obtain the continuous semantic features of the movie, despite a 2-s temporal resolution, we smoothed the text information in a sliding window approach. In other words, we created a smoothing window of 4 min by calculating the averaged bag-of-words vector corresponding to the center timepoint based on the sum of the bag-of-words vector of timepoints within the surrounding 4 min (2 min before and 2 min after a given point in time). The window size of 4 min was selected by a systematic grid search procedure (Supplementary Fig. 10). Indeed, we found that the 4-min window length can maximize both the average and maximum correlation between brain states' presence and extracted semantic contexts. Additionally, we conducted a quantitative comparison of the average link strength between the DN group and the whole group (Supplementary Fig. 13). Four minutes was determined to be the optimal choice. Based on these two pieces of evidence, we decided to use a window length of 4 min.

Subsequently, we benefited from a trusted NLP technique[29], term-frequency inverse-document-frequency (tf-idf), to re-represent the two text word count matrices $M_{sub}$ and $M_{des}$ tracking events of the 2-h movie. Tf-idf is a metric to transform the bag-of-words counts into word frequencies so that the word's global prevalence in our entire movie was appropriately considered. Tf-idf consists of two parts, term frequency (tf) and inverse document frequency (idf). The tf component reflects the similarity of a word's effect in the corpus. The idf component is defined as the logarithmic form of the inverse fraction of the size of the corpus.

$$\text{tfidf}(w,t) = \text{tf}(w,t) \times \text{idf}(w,t) = \frac{f_{w,t}}{\Sigma_{w' \in t} f_{w',t}} \times \log\left(\frac{T}{1+n_{w,t}}\right), \quad (1)$$

where w is the word corpus entry, $t$ is the timepoint, $T$ is the total number of timepoints and $n_{w,t}$ is the number of times points whose bag-of-words vector contains the word w. Consequently, a higher tf-idf value directly represented a higher salience of the word at one timepoint. At the same time, the salience of the word was calibrated by the abundance of that word in the general movie text corpus. As a result, the transformed word count vector derived, using sliding-window expansion, at each timepoint encapsulated the rich structure of the movie.

To automatically search through the space of candidate semantic representations which point to a similar contextual meaning, latent semantic analysis (LSA) was a natural choice of NLP technique[29]. LSA naturally assumes that words with similar meaning co-occur in the text i.e., the so-called distributional hypothesis[35]. Therefore, by applying LSA to our corpus of preprocessed movie subtitles and descriptions separately, we extracted two unique sets of semantic contexts that proxy underlying movie events. We extracted the $s$ = 200 semantic contexts from the word count matrices $M_{sub}$ and $M_{des}$. The full-length movie's text content was thus decomposed into a $k$ dimensional vector.

In the traditional formulation, LSA performs a singular value decomposition (SVD) of the sparse rectangular word count matrix M. The resulting semantic contexts are ordered based on the explained variance. This implicit property of ordered importance is only suitable

for a smaller s. While the value of the s increases, the first several components account for the bulk of the semantic variation about the movie events, while the other components are more nuanced and contribute less to explaining parts of the movie narrative. As the level of content information carried by semantic components decreases, empirically, their Pearson's correlation with brain states also become weaker. To maintain evenly distributed semantic contexts which decompose the full-length movie, we added a non-negativity constraint to the traditional SVD, which takes the form of non-negative matrix factorization (NMF):

$$M = W \times H, \text{ with } W \geq 0, H \geq 0, \quad (2)$$

where the matrix $W \times H$ is an s-rank approximation of the original word count matrix $M$. The matrix $H$ contains orthonormal columns representing word weights, by which we interpreted the events embedded in the semantic context. In doing so, NMF-LSA detects and extracts evenly distributed collections of semantic themes. The non-negative estimated parameter values of each column in the low dimensional projection matrix W indexed the importance and presence of the given semantic context shifts across the movie timeline. On the other hand, our NMF-LSA can be viewed as modeling the underlying movie events (indicating a solitary plot or character) of the evolving movie narrative.

In fact, the choice of an optimal number of semantic contexts s affects the downstreaming results only weakly. In the following correlation analysis with brain states, we selected the top n correlation links out of $s$ candidate semantic contexts as the indicator of strength of semantics-brain associations. Therefore, the screened dimensionality of $s$ is intrinsic to the scope of our analyzed movie and was empirically determined from the data themselves. In the present investigation, a larger s will generate repetitive and trivial semantic contexts (since the total number of underlying movie events is fixed) instead of complete semantic contexts, and a lower s cannot offer extensive sets of movie events covering the whole movie content. In this way, we identified 200 as a sweet spot for the total number of approximated movie events to balance the integrity and expressivity of the generated semantic contexts.

**Hidden brain state patterns via hidden Markov modeling.** To pair the movie narratives with delineated coupling patterns of brain dynamics at subregion resolution, we have turned to hidden Markov models to derive a sequence of underlying functional coupling brain states, separately in each subject. The core assumption behind HMM is that the data distribution of the observed timepoints is emitted from a sequence of to-be-uncovered hidden states, where this sequence of hidden states would switch and recur based on time-invariant transition probabilities. The complex dependencies between hidden states are simplified by a model specification, whereby the current timepoint is conditioned on the previous timepoint (i.e., the first-order Markov assumption). More formally, the probability of state $j$ being active in the current timepoint t is determined by which state was active at the previous timepoint $t-1$:

$$\Pr(S_t = k) = \sum_l \Theta_{l,k} \Pr(S_{t-1} = l), \quad (3)$$

where $\Pr(\dots)$ represents the probability of a hidden state to be present, $S_t$ is the hidden state at timepoint $t$, $S_{t-1}$ is the hidden state at timepoint $t - 1$, $\Theta_{l,k}$ gives the transition probability from previous state l to current state $k$.

Previous brain-imaging investigations endorsing HMMs usually concatenated all subjects' time series to form a group-level set of hidden states. In our analysis, given the unusual abundance of >3000 timepoints per subject, compared to previous naturalistic movie modeling studies, we were able to model the individualized network

dynamics. Hence, a separate collection of HMMs was estimated for each subject. Moreover, we wanted to identify the direct linkage between limbic subregions and neocortical networks. Therefore, for each subject, we examined 14 designated region-network combinations, with a total number of features

$$R = r_{network} + 2\, r_{limbic}, \tag{4}$$

where $r_{network}$ carries the number of regions from the Schaefer-Yeo networks entry and $r_{limbic}$ carries the number of principal component (PC) embedding expressions of either HC or AM for one hemisphere. First, the dimension of voxels averaged regions $r_{network}$ ranged from 5 to 24 across seven Schaefer-Yeo networks. Second, we extracted PCs for a given limbic target region: HC or AM. For each limbic target region, we used one PCA model instance to estimate $r_{limbic} = 3$ identical PCs (i.e., the two hemispheres shared the same set of PCA-derived singular vectors). Specifically, we concatenated the functional activity signals of 19 (HC) or 9 (AM) subregions for each hemisphere along the time axis so that we could train one uniform PCA model per limbic target region. In this way, we allowed for the possibility of lateralization effects of the limbic regions. After PCA, both HC and AM subregions were embedded into $2\, r_{limbic} = 6$ PCs. For each timeseries data matrix $X_{I,c}$ ($T \times r$) of a certain subject i and one region network combination c, we used the multivariate Gaussian as the observational model. More specifically, we modeled the dependency of observed data with the hidden state as:

$$x_t | S_t = k \sim \text{Multivariate Gaussian}\left(\boldsymbol{\mu}_k, \boldsymbol{\Sigma}_k\right), \tag{5}$$

where we denoted observation data vector $x_t$ as one timepoint slice in timeseries data matrix $X_{I,c}$, $S_t$ as the the hidden state at the timepoint $t$, the active state index as $k$, $\boldsymbol{\mu}_k$ as the mean vector of the multivariate Gaussian distribution, representing mean voxel-averaged BOLD signals corresponding to the constituent ROIs of each input region-network combination, $\boldsymbol{\Sigma}_k$ as the covariance matrix of subregion-subregion effects of the region-network combination. Our observation model encapsulated the assumed data distribution of each hidden state $k$ based on the parameters $\boldsymbol{\mu}_k$ and $\boldsymbol{\Sigma}_k$.

The Python package hmmlearn (https://hmmlearn.readthedocs.io/en/latest/) was used to perform the model parameter estimation based on an expectation–maximization algorithm. The number of training steps was set to 500 iterations. Notably, solving the HMM problem using the solver algorithm poses a non-convex optimization problem. This means that in one model estimation the converged HMM solution could fall in a local minimum. So for each unique model training iteration, with the identical input data (i.e., given a region-network combination), we performed 100 model estimation instances based on distinct randomized starting parameter values to obtain 100 different HMM instances for one specific training setting, from which we selected the one with the largest average Pearson's correlation strength with the semantic components. In particular, to obtain the subregion contributions of the two limbic regions, we projected the PCs back to the ambient anatomical space of the subregions. An HMM mode also contains the temporal characteristic for one state (cf. the section entitled Temporal analysis across different HMM solutions). Four patterns of one HMM training setting forms a coherent description of functional couplings, explaining the multi-modal neural activity for one specific region-network combination of one subject (Supplementary Figs. 3–5).

Typically, in many data analysis scenarios, the training of an HMM instance cannot afford high-dimensional input data for the total number of model parameters to be estimated increase with the number of features at a rate of $O(n^2)$. Because of our uncommon training approach (i.e., train an HMM solution corresponding to each region-network combination separately), we addressed the model complexity concern of the HMM modeling. This modeling agenda also carefully

aligned our estimated dynamic patterns with the biological questions of comparing different levels of the neural processing hierarchy.

Finally, for the purpose of model selection, we carried out a principled procedure to choose a useful number of HMM states. We rigorously evaluated four complementary metrics for selecting the optimal number of hidden states, which nominated $n = 4$ as the optimal solution. First, we calculated the strength of semantic context-brain association patterns (Supplemetary Fig. 1A). Specifically, charting the HMM solutions across choices of 2–8 hidden dynamic brain states, we compared the mean value of the top 10 Pearson's correlation coefficients between HMM models' state presence and 200 extracted semantic contexts. At the number of four states, this metric reached its peak value which represents the strongest brain activity and movie narrative correlation. Second, we modeled subject level variance by summarizing the mean Pearson's correlations across 15 subjects' HMM state presence probabilities (Supplementary Fig. 1B). Based on the goal of reflecting idiosyncrasies, we selected a lower value such that more discovered dynamics could be extracted. After four states, this value stabilized. For the third and fourth criterion, we evaluated the Bayesian information criterion (BIC) under the condition with limbic subregions and without limbic subregions (Supplementary Fig. 1C, D). BIC measured the likelihood of the model. We prefer a lower value. Whether limbic subregions are present or not, BIC reached its global minimum at four states. Overall, four brain states better reconcile brain-movie associations, subject level dynamics, and quality of model fit.

## Subject-level associations between movie events and dynamic brain states

Our primary analysis sought to find which layer of the neural processing hierarchy best tracks semantic contexts across the evolving movie narrative. Therefore, we performed a comparison of association strengths between brain states and semantic contexts, where each model directly reflects the semantics-brain correspondences of the isolated region-network combinations. As per the hidden Markov model, the presence vectors ($T$ total timepoints $\times 1$) represents the probability of each state along the time axis. The contained probability values of four states at one given timepoint adds up to one. On the other hand, matrix $W$ ($T \times s$) generated from NMF-LSA (cf. natural language pipeline section) provided s different semantic components with the projected embedding expression of the word count matrix of the movie. Subsequently, we computed the pairwise Pearson's correlation strengths between $s = 200$ semantic contexts embeddings and $n = 4$ brain states. A Pearson's correlation matrix $P_{semantic}$ ($s \times n$) for semantic contexts thus reflected how brain states correlated with a large collection of semantic contexts. The same procedure was also performed for human curated annotations, where another Pearson's correlation matrix $P_{annotation}$ ($52 \times n$) for annotations depicts how brain states are linked with hand selected descriptions of the narrative.

For each state, we computed the average value of the top 10 association values as the aggregation method for one state's unique semantics-brain link strength. Then, we summarized each of the seven Schafer-Yeo networks' parcel-wise average neural activity responses across two different limbic partners (Fig. 4). Finally, $p$ values were estimated to assess the significance of group differences between the DN's value and the other six networks' average value under a two-sample t-test.

## Temporal analysis across different region-network combinations

To analyze the temporal characteristics of our derived brain states in greater detail, we calculated the dwell times of each HMM state, which refers to the duration of time that a given state is visited. To compute the dwell times, we adopted the same method used in previous studies[113]. Specifically, we assigned each timepoint to the state with the highest probability and calculated the duration of time that each state

was visited. We then aggregated the dwell times at the single subject level by comparing the average and volatility across four states for the same HMM model. This allowed us to compare the processing timescale differences across 14 region-network combinations. Following this, we averaged the temporal characteristics across 210 estimated HMM instances (15 subjects × 14 region-network combinations). In doing so, we were able to directly characterize how the temporal processing characteristics varied across subjects and functional network level dynamics.

### Group-level association analysis of brain signatures with external descriptions

To elevate the subject-level patterns to the group level, we designed an analysis framework combining HMM state presence information and PLS-R. We used the probabilistic state presence information to segment the whole movie timeline into four partitions, each one associated with one of the four dominant brain states. To align the state indices across different subjects, we adopted the Hungarian algorithm[114] to reorder the four brain states across 15 subjects. We selected subject 1 as the template to be matched against. We aimed to minimize the distance between the four brain states' observation model parameters for subject 1 and each of the other 14 subjects. In so doing, the brain states were reordered, which means, for example, brain state 1 points to a similar representation across the 15 subjects. We then concatenated the partitioned neural activity responses for each of the four states and generated four group-level segmented neural responses matrices $G_k$ where $k \in \{1, 2, 3, 4\}$

We then inferred the dominant PLS-R direction for each of the four $G_k$ to provide insights into how external descriptions in the movie can explain functional variability in four different brain states. We concatenated the semantic context embeddings $W$ and human curated annotations A into a combined external descriptions matrix $E$ with the dimension of $T$ total timepoints $x$ ($s$ semantic contexts + 52 annotations). We then used the same state-specific partition of across-subject concatenated external descriptions matrix $E$ into four group-level external descriptions matrices $E_k$ where $k \in \{1, 2, 3, 4\}$. PLS-R was a natural choice of method to find the single most explanatory signature connecting neural responses and external descriptions. For each state $j$, we trained an independent PLS-R model by implementing the method from the Python package sklearn. The input variable set comprised a state's external descriptions matrix $E_k$, while the target variable set comprised the partitioned neural responses $G_k$. In our study, the one-dimensional embedding of both $E_k$ and $G_k$ represented the most explanatory projection that maximized the covariance between contextual information of the ongoing movie narrative and the sets of brain subregion activities. Concretely, the component parameter values corresponding to each of the two sets reflected relative contributions to maximize the semantics-brain relationship. For the PCs of limbic subregions (AM and HC), the PCA inverse transformation was used to map the parameter values of the PCs to the original limbic subregion level, where we identified the functional interconnection of neocortical subregions. On the descriptions' side, we identified the most prominently contributing semantic contexts and annotations among three categories (place, emotion, and rest) from parameter values corresponding to $E_k$ loading to nominate properties of the tracked movie semantics and the neurocognitive categories at play. Collectively, the four cross-associations of corresponding brain subregions and top contributing descriptions formed four coherent signatures that may speak to the multi-mode mechanism of DN circuits paired with both HC and AM.

To test the generalization ability of PLS-R model, we performed a 20-fold cross validation. Instead of splitting the subject-wise concatenated data matrix, it is more rational to divide each subject's state specific time sequence into 20 folds. Then, we concatenated the 19 folds across subjects to offer the training set to for PLS-R. The left out onefold

data were also concatenated across subjects to offer the testing set. The PLS-R model was trained on the training set with the same procedure. Subsequently, the Pearson's correlations were calculated between the two one-dimensional embeddings on the testing set. Then, this process was repeated twenty times so that all 20 folds data were covered. The summarized results were shown in Supplementary Fig. 8.

### Reporting summary

Further information on research design is available in the Nature Portfolio Reporting Summary linked to this article.

## Data availability

The intermediate data generated in this study have been deposited in https://osf.io/6s2xh/. The raw fMRI data, structural MRI and annotations are available at studyforrest's official website https://www.studyforrest.org/. Source data are provided with this paper.

## Code availability

Code is available here: https://github.com/dblabs-mcgill-mila/hmm_forrest_nlp.

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

## Acknowledgements

D.B. was supported by the Brain Canada Foundation, through the Canada Brain Research Fund, with the financial support of Health Canada, National Institutes of Health (NIH R01 AG068563A, NIH R01 DA053301-01A1, NIH R01 MH129858-01A1), the Canadian Institute of Health Research (CIHR 438531, CIHR 470425), the Healthy Brains Healthy Lives initiative (Canada First Research Excellence fund), Google (Research Award, Teaching Award), and by the CIFAR Artificial Intelligence Chairs program (Canada Institute for Advanced Research).

## Author contributions

All authors contributed extensively to this work. E.Y., and D.B. conceived and designed the study. J.K., A.J.H., and G.D.M. provided guidance on data analysis and interpretation. E.Y. and F.M. performed the data analysis, with input from J.K., A.J.H., and G.D.M. B.M. and E.S.F. contributed to the development of the methodology and provided critical feedback on the manuscript. E.Y., and D.B. wrote the manuscript, with input and revisions from all authors. All authors read and approved the final manuscript. D.B. led the data analysis.

## Competing interests

D.B. is a shareholder and advisory board member at MindState Design Labs, USA. Other authors declare no competing interests.
