## [Peer Review File · Nature Communications]

The default network dominates neural responses to evolving movie storiesReviewer #1 (Remarks to the Author):

This is a very interesting study, I very much enjoyed reading the manuscript through most enthusiastically. I personally think that the results and methods that were developed have the potential of significantly contributing to the research field. I have a number of comments that I hope will help the authors increase the scientific quality of their manuscript.

1) The studyforrest dataset is a good one, however, it is dubbed in German (as the subjects were all native German speakers), the number of subjects is only 15. This raises a question on to what extent the results can be generalized. In the data resources section it is stated that 14 out of these 15 subjects had seen the movie before, and that they all were recruited for audiovisual and audio-only versions of the study, however, I understood that here only the audiovisual data were processed? The strength of the dataset is that there are multiple annotations that the authors utilized here to analyze how the semantics in the movie relate to brain activity, in addition to the language content derived from the time-aligned subtitles. However, given the shortcomings, I am wondering if the authors could find some additional freely accessible dataset (e.g., from amongst the datasets recently made open-access from the Hasson lab via Scientific Data in 2021) using which they could try to verify the results that they obtained with the studyforrest set. This in my mind could significantly increase and significance and replicability of the results of this excellent study that the authors have carried out.

2) Since the default mode network emerges as central player in processing of the semantics in the movie, I came to think of works by Nguyen et al. Neuroimage 2019; Yeshurun et al. Psychol Sci 2017, these and others were reviewed recently by Jaaskelainen and colleagues in Neuroimage in 2021. The relevant studies could be cited where appropriate, also I came to think an early study by Iacoboni et al. Neuroimage 2004 as relevant as it showed DMN activity during watching of social interactions. Lastly, work by Moshe Bar on the semantic-associative processes run by DMN coupled with hippocampus is something that might be relevant for the authors to link.

3) One notable omission in literature cited by the authors is the work from Jack Gallant's laboratory, as they have modeled brain activity based on semantics in movies and narratives e.g. papers by Huth et al. in Neuron 2015 and Nature 2016, and other relevant work. Citing these works would allow placing of the current findings better into context and thus also highlight the unique contributions to scientific knowledge.

4) On the inter-individual differences that the authors describe at brain level, the studyforrest dataset unfortunately does not allow for cues on individual semantics (e.g Saalasti et al. Brain Beh 2019). While I applaud the approach overall, I see this as an essential shortcoming that the authors might wish to mention briefly, to guide follow-up studies inspired by this study. Also, some of the other datasets, as suggested above in #1, might have this type of information and thus could be tried out.

5) The methods in places are difficult to follow. I suggest the authors spend some time with for example a colleague who has not seen the manuscript before to spot the places where it is difficult to follow. Increasing clarity throughout would in my opinion help enhance the potential impact of this study.

6) I find it somewhat perplexing that there are only 4 brain states that are discovered. Are these perhaps specific to this movie, which is of Comedy genre?

7) In the illustrations the labels are in places with awfully small font, this should be improved somehow.

Reviewer #2 (Remarks to the Author):

The study of Yang et al. extends upon previous work modelling the fMRI signal dynamics elicited by a movie by adopting machine learning to define the semantic context of movie scenes. The

approach allowed estimating the likely occurrence of several subject-specific 'region-network combinations' during movie watching that link with information of the evolving narrative (semantic contexts). One of the key findings of this study is the core role of the default mode brain network in tracking and integrating the unfolding storyline.

Overall, the paper is clear. That said, the contribution of this work relative to the existing literature appears incremental. For example, it has been shown that the processing of complex semantic information involves brain regions comprising "slower" high-order networks, particularly the default mode (e.g., works from Hasson and Norman labs). After reading the paper, I was unsure about what new knowledge about brain function was gained. Moreover, I have several methodological concerns (below) and the significance of the findings is arguably overstated (e.g., delineation of core mechanisms supporting narrative-specific inter-regional coupling).

In the context of the above general impression, I also have the following comments:

1. The authors state that: "our analytical framework was carefully tailored to capture rich information in the full-length 2-hours movie". It is thus surprising that four hidden brain states represent the optimal resolution, across region-network combinations and subjects. This resolution is coarse (e.g., Vidaurre et al., 2017; van der Meer et al., Nat Comm, 2020) and likely to preclude the detection of meaningful (e.g., region-network or subject-specific) movie-related dynamics. Related to this, I am not convinced that the selection of the optimal number of hidden states should be guided by the correlation between brain activity and movie narrative. This approach may bias the results towards "high order" region-network combinations (see also my comments below). Would it not be more principled to estimate hidden brain states and then assess how such states link to semantic contexts across subjects?

2. The activity of various region-network combinations during movie watching is likely interlinked. Therefore, I struggle to understand the rationale of only independently assessing brain states in region-network combinations. This approach may obscure critical information on movie-induced whole-brain state dynamics, arguably necessary to interpret region-network-specific fluctuations linked to semantic contexts.

3. The downsampling of text information from 2 seconds to 4 minutes was motivated by a systematic grid. However, the values presented in SFigure 10 are pretty similar between some window lengths. Moreover, the selected window length facilitates the detection of default mode activity (e.g., figure 4 in Baldassano et al., 2017) but may obfuscate the detection of potentially relevant dynamics in "lower" region-network combinations that occur on shorter time scales (e.g., Vis&AM).

4. While I can appreciate the general interest in assessing limbic-neocortical activity patterns, the rationale for such analysis is unclear. Why are the amygdala and hippocampus considered key neocortical partners in this context? These two brain regions are known to have different levels of association with regions defining canonical brain systems (e.g., activity in the hippocampus has been linked with the activity of core default mode regions, including the angular gyrus and the posterior cingulate). Would these differential functional associations bias the results? If so, how (e.g., increased sensitivity to detect default mode dynamics)?

Reviewer #3 (Remarks to the Author):

Summary of the Paper

This paper sought to study naturalistic responses of humans (15 subjects) to watching the movie *Forrest Gump*, a 2-hour movie. To do this, processed fMRI recordings were paired with natural language descriptions of the movie, namely via a set of augmented subtitles and audio descriptions of the movie. Then, natural language embeddings were constructed from bag-of-words embeddings via a tf-idf transformation followed by non-negative LSA, classical NLP techniques. The result was 200 semantic contexts, which means a matrix $R^{T \times 200}$, where each of the 200 dimensions corresponds to a different aspect of the movie. On the fMRI side, PCA

was applied to the voxels composing the hippocampus and amygdala over time separately, yielding 6-dimensions corresponding to the limbic regions. These were appended to the voxel regions defined by 14 target pre-defined regions of the brain from the Schaefer-Yeo atlas. Then, an HMM model with 4 hidden states with a non-isotropic Gaussian emission model was estimated for each of the 15 subjects and each of the 14 brain regions. For each of the 14x15 combinations, 100 HMMs were fit with different starting seeds and the resultant HMM whose vector of hidden state probabilities over time (in R^T) best correlated (on average?) over the 200 T-dimensional contexts.

Given this setup, the goal was to better understand the dynamics of brain activity resulting from natural movie stimuli in different subregions of the brain, with special attention to the default mode network (which has been previously shown to have various properties connected with natural stimuli), the hippocampus, and the amygdala.

The paper then associates the estimated latent states of the HMM for each viewer with the 10 most closely related semantic dimensions via correlation analysis, and conducts an analysis of means and variances of time spent in these (now interpretable from a descriptive point of view) hidden states.

Overall, the claimed contributions were as follows:

- Provided a simple method for interpreting the hidden states of naturalistic fMRI data in terms of accompanying narrative text, and conducted studies of the dynamics of various brain regions and their connection w/the text.
- Lower level parts of the brain like the visual network had fitted hidden states with shorter dwell times and smaller variance
- Default mode network regions had fitted hidden states with dwell times with higher variance.
- Different hidden states of the HMM tracked different parts of the narrative, as measured by the semantic context states (e.g., each hidden state correlated with different sparse sets of semantic context dimensions). (Though this doesn't seem surprising since each HMM was chosen to maximize correlation with the semantic context vectors, and the correlations are not that strong ($r \sim .2$)).
- Functional correlation analyses of the default mode network, and the amygdala demonstrated that anatomically adjacent subregions of both exhibited coupling behavior that was the same in both
- Another demonstration of the fact that the default mode network appears to capture a lot of the brain activity involved in narrative processing in a variety of ways
- Dwell times of states varied a lot across subjects
- Lower level networks had timescales that were less volatile across subjects.
- Several fine-grained coupling signature pairings for different subregions of the brain.
- Analysis finding an association between the processing of emotional scenes with the amygdala subregions.
- Findings of lateralization in the amygdala which correlate with similar lateralizations in neural responses in emotion and work linkage experiments.

Assessment Summary

Overall, the paper applies simple approaches from machine learning to study naturalistic fMRI narrative data for a long movie. In particular, it appears to be the first work (as far as I can tell) to associate text descriptions with hidden states found via fitting an HMM, and to then conduct dynamics analysis of the resulting fit.

The main conclusion seems to be that the default mode network is relevant in narrative processing, but this conclusion has been established in several existing papers already (see survey [1]). However, there are several more fine-grained studies which examine dynamics properties and fine-grained function correlations between different subregions of the brain. Also of interest (and novel attention) is the focus on coupling behavior between the default mode network and different parts of the limbic region, the amygdala and the hippocampus, but while there were differences (as elaborated on pg. 14), it was hard to extract clear takeaways and the paper would

benefit from more clarity in this direction, particularly in the story of the paper.

There were also a few issues with the positioning of the paper - there are many highly relevant papers that the literature review missed, which should be added — in particular, many works for a long time have made connections between NLP methods and fMRI data, and sometimes of more sophistication than the methods used here, also on naturalistic, long narrative data (movies, books, etc). So while it does appear that the specific combination of applications of methods is new (HMM to analyze the dynamics of the fMRI data paired with semantic analysis for paired text), it is a stretch to say this is a big innovation. Additionally, the specific methods on each side (the HMM, and the text embeddings) are very simple and have been superseded by other approaches in various cases (see several of the missing papers from the literature review for more recent approaches to applying NLP in the context of fMRI data). On the HMM side, it seems relevant to consider HMM variants that have been specifically tuned and adapted to the problem of event detection in fMRI data [2].

Finally, there were several issues with the presentation - the sentences could be edited for verbosity and clarity, and overall the text would benefit from less purple prose. I also found the descriptions of some of the methodology hard to parse — the paper would benefit from more explicit descriptions with equations, and better notation. I also thought the connections to understanding consciousness were overreaching.

In general, I felt that the message of the paper was not clear enough. It could be greatly strengthened by more directly using the NLP components of the analysis to more concretely and strongly analyze the differences between the amygdala+DN and hippocampus+DN combinations. If a more thorough analysis was developed and applied to this part of the paper, then paper would be significantly better. It seems possible to fix this issue without too much additional effort — the conclusion in the last paragraph on pg. 16 is not well supported enough, and the characterization on pg. 17 is too loose and non-rigorous to make stronger claims. It would also benefit the paper to highlight this directional point in the beginning of the paper, rather than focusing on how NLP has not been applied to long narratives before (which is false), and also to avoid the emphasis on the finding that the default mode network plays a very strong role in narrative processing, which is not a novel finding (though it is always nice to see more evidence).

[1]: The default mode network: where the idiosyncratic self meets the shared social world. Yaara Yeshurun, Mai Nguyen, and Uri Hasson

[2]: Discovering event structure in continuous narrative perception and memory. C. Baldassano, J. Chen, A. Zadbood, J.W. Pillow, U. Hasson, K.A. Norman.

Some Example Missing Citations

[3]: Narrative event segmentation in the cortical reservoir. Peter Ford Dominey

[4]: Mapping neural activity to language meaning. L. Wehbe, A. Fyshe, T. Mitchell (and many, many previous works by these authors)

[5]: Aligning context-based statistical models of language with brain activity during reading. L. Wehbe, A. Vaswani, K. Knight, T. Mitchell

[6]: Mapping between fMRI responses to movies and their natural language annotations. Kiran Vodrahalli, Po-Hsuan Chen, Yingyu Liang, Christopher Baldassano, Janice Chen, Christopher Honey, Uri Hasson, Peter Ramadge, Kenneth A. Norman, Sanjeev Arora.

[7]: Decoding the Semantic Content of Natural Movies from Human Brain Activity. Alexander G. Huth, Tyler Lee, Shinji Nishimoto, Natalia Y. Bilenko, An T. Vu, Jack L. Gallant. (And many many previous works by these authors).

[8]: Low-dimensional Structure in the Space of Language Representations is Reflected in Brain

Responses. Richard Antonello, Javier S. Turek, Vy Vo, Alexander Huth.

[9]: Bayesian Surprise Predicts Human Event Segmentation in Story Listening. Manoj Kumar, Ariel Goldstein, Sebastian Michelmann, Jeffrey M. Zacks, Uri Hasson, Kenneth A. Norman

[10]: Reconstructing the cascade of language processing in the brain using the internal computations of a transformer-based language model. Sreejan Kumar, Theodore R. Sumers, Takateru Yamakoshi, Ariel Goldstein, Uri Hasson, Kenneth A. Norman, Thomas L. Griffiths, Robert D. Hawkins, Samuel A. Nastase.

[11]: Narratives: fMRI data for evaluating models of naturalistic language comprehension Nastase, S.A., Liu, Y., Hillman, H., Zadbood, A., Hasenfratz, L., Keshavarzian, N., Chen, J., Honey, C.J., Yeshurun, Y., Regev, M., Nguyen, M., Chang, C.H., Baldassano, C., Lositsky, O., Simony, E., Chow, M.A., Leong, Y.C., Brooks, P.P., Micciche, E., Choe, G., Goldstein, A., Vanderwal, T., Halchenko, Y.O., Norman, K.A., & Hasson, U.

Questions

1. How exactly do you label the 200 semantic dimensions with names beyond looking at word clouds? How did you associate them with emotions? I didn't quite understand the part about the 52 annotations. (In particular, what exactly is being annotated? The word clouds from the semantic vectors? The movie scenes? The text of the movie? The reactions of the viewers?)
2. How does your HMM fit compare to event segmentation HMM of Baldassano et. al. [2]?
3. How exactly do you calculate dwell times from the estimated probabilities of states? Do you pick the max probability per time point and assign that state as the label? I missed it if this was stated somewhere.
4. If you use the full trajectory of semantic contexts and HMM fits to choose the best HMM fit, you are using information from the whole movie. This is a problem when you do your held-out regression analysis with partial least squares, since the fitting of the unsupervised step used information from the whole movie.
5. On pg. 9, are the default mode network regions anatomically adjacent to each other? What is the relation to the amygdala subregions? The language is not clear.
6. It would be good to quantify lateralization effects rather than describe them; I felt the description was imprecise and do not amount to an argument for explaining "links between the brain and movie contextual information" - at least I don't see how this conclusion follows.
7. In Fig. 4, could you include a pointer to the section of Methods you're referring to? Also, it's not clear to me why the box-plots in Fig. 4 for regions other than the limbic region are not identical across amygdala and hippocampus — I didn't completely understand the methods section description of what Fig. 4 means. Is it because of the fact that you concatenate either the amygdala or hippocampus 3-dim PCA embedding to the brain regions, and this affects the fit of the HMM? If so this wasn't clear from the description.
8. How similar are the 4 "brain-states" (hidden states of the fit HMM) across the 210 total HMMs? It's not clear that these should be similar, right?
9. How do you account for the variation in the HMM fit? It seems to me that it's plausible that after you pick the HMM fit with maximal correlation to the semantic contexts over 100 initializations, the fact that you are still doing 210 HMM fits (and then average over 14 of them per person) suggests that there could be significant variance in the estimated states due to the fitting process and thus the estimated dwell time - it's not clear that this is only due to the subjects being different. Can you do a statistical test over the variation of the HMM fitting process to check the significance of the standard deviation variations over each human ?
10. Fig. 6 and Fig. 7 would be much clearer to understand if you wrote down the partial least squares objective. From a visual perspective, it might also be useful to have Figs. A and B on the left and Fig. C on the right, both with arrows pointing to Figure D below them.
11. For the results on pg. 14, it would be nice to have a quantitative summary of the associations, at least for the main points you want to make (e.g. amygdala and emotions).

Style Issues and Grammar etc.

- * Equations should be clearly written out. It would be helpful to be clearer about the dimensions of various matrices and vectors, should explicitly add the optimization problem formulations, etc.
- * For instance, on pg. 13, the description "extract the dominant signature that tracks how the 200 semantic context expressions..." is very confusing to read. What is a "signature?" It would be much better if the PLS-R objective was just written out and the goal more clearly described.
- * As an example of word choice issues - on pg. 13, "triangulated" is an odd choice of word to use - maybe "associated".
- * It would be better to avoid using "*" for denoting matrix shapes, better is "x" or "\times".
- * There are some odd tense choices throughout (past tense a lot)
- * There is a lot of passive voice that could be removed.
- * There is a lot of unnecessary verbiage and unnecessary adjectives.

Reviewer #1 (Remarks to the Author):

This is a very interesting study, I very much enjoyed reading the manuscript through most enthusiastically. I personally think that the results and methods that were developed have the potential of significantly contributing to the research field.

We are extremely grateful to the reviewer for this positive assessment of our work.

I have a number of comments that I hope will help the authors increase the scientific quality of their manuscript.

1) The studyforrest dataset is a good one, however, the number of subjects is only 15. This raises a question on to what extent the results can be generalized. In the data resources section it is stated that 14 out of these 15 subjects had seen the movie before, and that they all were recruited for audiovisual and audio-only versions of the study, however, I understood that here only the audiovisual data were processed? The strength of the dataset is that there are multiple annotations that the authors utilized here to analyze how the semantics in the movie relate to brain activity, in addition to the language content derived from the time-aligned subtitles. However, given the shortcomings, I am wondering if the authors could find some additional freely accessible dataset (e.g., from amongst the datasets recently made open-access from the Hasson lab via Scientific Data in 2021) using which they could try to verify the results that they obtained with the studyforrest set. This in my mind could significantly increase and significance and replicability of the results of this excellent study that the authors have carried out.

We thank the reviewer for this valuable point. We are aware of the value of validation in different scenarios. To this end, we have carried out several different validation experiments to ensure the robustness of our findings and conclusions. However, we would like to emphasize that the studyforrest dataset is unique for its continuous fMRI recording length and its richness of different kinds of metadata (e.g., annotations). In short, our dataset has one of the most in-depth functional brain scans in the naturalistic neuroscience field. This focus on within-subject detail, not quantity, is why fewer subjects were scanned, but each participant yielding much more overall information. We provide more details on this reasoning in the following:

Firstly, in the studyforrest resource, each subject has 2-h length brain scanning with neural activity responses covering the whole movie. We took advantage of this unique dataset opportunity to train subject specific HMM models: a modeling opportunity that, to our knowledge, has not been available to the imaging neuroscience community before. Benefitting from the opportunity to estimate subject specific HMM models, we are able to

model how each individual and each region-network combination responds to the movie material. Along with other external descriptions, we offered a full, faithful picture of each subject's individual functional coupling dynamics. In contrast, other datasets typically contain a 10-20 min long scan during movie watching (3-56 min in the dataset the reviewer kindly mentioned, see Fig below). Therefore, our specially designed subject-level pipeline is hard to replicate in other movie datasets.

Secondly, in our study, we introduced an unprecedented microanatomical depth in naturalistic movie modeling by incorporating subregion definitions of the amygdala and hippocampus (together >50 subregions) to examine their functional coupling dynamics with the cortical networks. To achieve this unprecedented precision in studied neural activity responses, we utilized T1 weighted images from the studyforrest dataset to derive subject-specific volume data for 18 amygdala subregions and 38 hippocampus subregions. Using capabilities of Freesurfer's computational anatomy suite, we generated individual limbic region segmentations, which were then utilized to extract functional activity features from fMRI to be analyzed in downstream analyses. This approach would be difficult to replicate without access to the structural images of the same subjects provided by the dataset.

Story	Duration	TRs	Words	Subjects
"Pie Man"	07:02	282	957	82
"Tunnel Under the World"	25:34	1,023	3,435	23
"Lucy"	09:02	362	1,607	16
"Pretty Mouth and Green My Eyes"	11:16	451	1,970	40
"Milky Way"	06:44	270	1,058	53
"Slumlord"	15:03	602	2,715	18
"Reach for the Stars One Small Step at a Time"	13:45	550	2,629	18
"It's Not the Fall That Gets You"	09:07	365	1,601	56
"Merlin"	14:46	591	2,245	36
"Sherlock"	17:32	702	2,681	36
"Schema"	23:12	928	3,788	31
"Shapes"	06:45	270	910	59
"The 21st Year"	55:38	2,226	8,267	25
"Pie Man (PNI)"	06:40	267	992	40
"Running from the Bronx (PNI)"	08:56	358	1,379	40
"I Knew You Were Black"	13:20	534	1,544	40
"The Man Who Forgot Ray Bradbury"	13:57	558	2,135	40
Total:	4.6 hours	11,149 TRs	42,989 words	
Total across subjects:	6.4 days	369,496 TRs	1,399,655 words	

Typical length of movie snippets that fMRI researchers analyzed before (Nastase et al., 2021)

Thirdly, the studyforrest dataset contains different collections of rich meta-information that are hardly available in other movie fMRI datasets. The large amount of carefully human-labeled annotations helped us to bridge the dynamic brain state effects at sub-region levels with intuitively well-known and human-interpretable psychological concepts (e.g. emotions such as fear). The rarely available annotations of our movie data also helped us to replicate previous knowledge on the primary neurobiology that was derived from single cell level animal studies in prefrontal and hippocampus pathways. For example, neurons in the CA1 have previously been reported to be sensitive to bright vs dark environments during experiments on light condition transitions in rodents (Quirk et al., 1990). The same sensitivity to ambient light conditions also emerged in our study by using the previously unavailable time of the day annotation available in the studyforrest dataset. Besides, the placement of the camera indicates spatial boundary information in the movie. This new annotation also allowed us to find that the subiculum potentially shares the same function of identifying spatial boundary in space as has previously been shown in rodents study (Barry et al., 2006; Hartley et al., 2000). Specifically, for the same movie content, except the original movie subtitles, the additional narrator descriptions from an audio-only version provides valuable text sources that we submitted to our analyses. The cross language and cross text sources validation that we carefully conducted bolsters the robustness of our findings and conclusions.

In addition to the cross language and cross text source validation, we made several efforts to rigorously validate our core conclusions from modelling. We summarized the validation results in Fig 2. Each box contains 15 subjects' individual semantic brain link strength values, and the default network (red) box always showed a larger median value (i.e., the central line in the box). Therefore, the conclusion that the default network is relatively more involved in the movie watching process is robust under several experimental settings. We tried different data compression methods, and projected semantic space dimensionality. Additionally, to eliminate the possibility that the strong semantic brain link of the default network is due to randomness, we also performed empirical permutation tests by shuffling the DN's time series 1000 times (Fig. 3). The statistical test also indicated significant findings (p values < 0.001). As a result, we showed empirical evidence for the robustness of our original results even in an across-subject setting although we fitted single subject HMMs.

Supplementary Figure 7: Functional coupling dynamics across different text extraction methods and text sources. We performed several confirmatory analyses supplementing the main text’s Fig. 4. These supplementary experiments aimed to analyze the robustness of the conclusion, that the DN tracks most of the semantic contexts. The top row showed the results paired with amygdala as the extra limbic region and the bottom one was with hippocampus. Inside the plot, each x tick denotes one unique sets of semantic contexts. The “des” represented that the text data is the human description of the movie instead of the subtitle (sub). The two different text extraction methods used for LSA (cf. methods) were non-negative matrix factorization (NMF) and singular value decomposition (SVD). We generated 200 semantic contexts via NMF-LSA and 5 semantic contexts via SVD-LSA. Across different sets of semantic contexts, the median values of DN’s semantic brain link strength were the largest.

Permutation test of the DN&AM and DN&HC semantic brain link strength.

We have added a new limitation section to acknowledge this helpful reviewer point. "It is important to acknowledge some limitations when interpreting our results and conclusions. Firstly, appreciation of findings from the StudyForrest dataset need to take into account the fact that the majority of the subjects had previously watched the movie and are not English native speakers. This circumstance could potentially color some of the obtained results. For example, the familiarity of the subjects with the movie could lead to a diminished surprise level."

Nastase, S. A., Liu, Y.-F., Hillman, H., Zadbood, A., Hasenfratz, L., Keshavarzian, N., Chen, J., Honey, C. J., Yeshurun, Y., & Regev, M. (2021). The "Narratives" fMRI dataset for evaluating models of naturalistic language comprehension. *Scientific Data*, 8(1), 250.

2) Since the default mode network emerges as central player in processing of the semantics in the movie, I came to think of works by Nguyen et al. *Neuroimage* 2019; Yeshurun et al. *Psychol Sci* 2017, these and others were reviewed recently by Jaaskelainen and colleagues in *Neuroimage* in 2021. The relevant studies could be cited where appropriate, also I came to think an early study by Iacoboni et al. *Neuroimage* 2004 as relevant as it showed DMN activity during watching of social interactions. Lastly, work by Moshe Bar on the semantic-associative processes run by DMN coupled with hippocampus is something that might be relevant for the authors to link.

We appreciate the mentioned papers and understand their core contributions as follows:

1. DMN activity reflects the people's own interpretation to narratives (Nguyen et al., Shared narrative interpretation is correlated with neural similarity in DMN and FCPN. Yeshurun et al., *Psychol Sci* 2017).
2. Jaaskelainen and colleagues in *Neuroimage* in 2021: Temporal-receptive windows and event-segmentation are vital short-term memory mechanisms.
3. Iacoboni et al. *Neuroimage* 2004 : DMN is linked with social interaction and social perception.

We thank the reviewer for bringing these papers to our attention. In light of their findings, we have made amendments to our introduction in response to this helpful reviewer point:

“Semantic processing has been proposed to be essential for human higher-order functions (Bar et al., 2006). Due to the brain’s specific energy and wiring constraints (Bullmore and Sporns, 2012), efficient high-level information integration and retrieval functions are needed to realize many types of advanced neural processes. To clarify the definition of semantics, the word “semantic” has been used in different fields, including semantic memory, semantic processing, and linguistics (Binder and Desai, 2011; Binder et al., 2009; Brandman et al., 2021; Bzdok et al., 2013b; Eickhoff et al., 2016). Here, the use of the term is more closely linked to the notion of “structured knowledge of the external world”, as proposed by Binder and colleagues (2009). We thus executed a survey of seven functional brain networks regarding the strengths of their semantic context-brain links. In so doing, we established that the DN’s neural responses were most intimately linked with momentary shifts in the semantic contexts throughout the whole movie. **In addition, DN activity has previously been reported to reflect the people’s own interpretation to narratives (Nguyen et al. 2017; Yeshurun et al. Psychol Sci 2017,) and to link with social perception (Iacoboni et al. Neuroimage 2004 :). Based on this previous progress, we argue here that it is important to obtain single subject-level resolution on such brain dynamics and offer a dedicated analytical toolkit and interpretation framework.”**

Nguyen, M., Vanderwal, T., & Hasson, U. (2019). Shared understanding of narratives is correlated with shared neural responses. *Neuroimage*, 184, 161-170.

Iacoboni, M., Lieberman, M. D., Knowlton, B. J., Molnar-Szakacs, I., Moritz, M., Throop, C. J., & Fiske, A. P. (2004). Watching social interactions produces dorsomedial prefrontal and medial parietal BOLD fMRI signal increases compared to a resting baseline. *Neuroimage*, 21(3), 1167-1173.

Yeshurun, Y., Swanson, S., Simony, E., Chen, J., Lazaridi, C., Honey, C. J., & Hasson, U. (2017). Same story, different story: the neural representation of interpretive frameworks. *Psychological science*, 28(3), 307-319.

3) One notable omission in the literature cited by the authors is the work from Jack Gallant’s laboratory, as they have modeled brain activity based on semantics in movies and narratives e.g. papers by Huth et al. in *Neuron* 2015 and *Nature* 2016, and other relevant work. Citing these works would allow placing of the current findings better into context and thus also highlight the unique contributions to scientific knowledge about the human brain.

We thank the reviewer for bringing this lack of precision to our attention. We have now added more references about works from Jack Gallant’s lab:

“NLP has become an increasingly valuable tool for studying the human language systems implemented in the human brain. Several studies have used advanced NLP techniques to build association maps between language features and brain activity. The diverse set of experimental tasks include predicting movie scenes (Vodrahalli et al., 2018), **detecting semantic selectivity (Huth et al., 2016b)**, and decoding semantic content (Fyshe et al., 2012; Hagoort, 2019). **In particular, the usage of NLP techniques to access semantic features of human language has been shown to be successfully mappable to brain activity before (Huth et al., 2016a; Huth et al., 2012).** The advent of large language

models, such as enabled by transformer architectures, has pointed to exciting features of the human brain, such as the possibility of shared computational design principles in form of next-world prediction mechanisms (Goldstein et al., 2022; Kumar et al., 2022a; Kumar et al., 2022b). Additionally, large open neuroscience datasets have emerged as a promising way to accelerate research in this area (Nastase et al., 2021). These developments are likely to continue to advance new insights into the relationship between language processing and neural processing systems.”

- Huth, A. G., Nishimoto, S., Vu, A. T., & Gallant, J. L. (2012). A continuous semantic space describes the representation of thousands of object and action categories across the human brain. *Neuron*, 76(6), 1210-1224.
- Huth, A. G., De Heer, W. A., Griffiths, T. L., Theunissen, F. E., & Gallant, J. L. (2016). Natural speech reveals the semantic maps that tile human cerebral cortex. *Nature*, 532(7600), 453-458.
- Huth, A. G., Lee, T., Nishimoto, S., Bilenko, N. Y., Vu, A. T., & Gallant, J. L. (2016). Decoding the semantic content of natural movies from human brain activity. *Frontiers in Systems Neuroscience*, 10, 81.

4) On the inter-individual differences that the authors describe at the brain level, the study forrest dataset unfortunately does not allow for cues on individual semantics (e.g Saalasti et al. Brain Beh 2019). While I applaud the approach overall, I see this as an essential shortcoming that the authors might wish to mention briefly, to guide follow-up studies inspired by this study. Also, some of the other datasets, as suggested above in #1, might have this type of information and thus could be tried out.

There is only one text annotation for the whole movie, so we cannot experiment with subject-specific reflections / feedback of the movie. The studyforrest dataset focuses on recording subject’s perception rather than collecting their subject-specific feedback, since this data collection initiative administered a passive viewing paradigm to the participants. Apart from the basic demographic and questionnaire of the movie content, no other feedback information was provided. This missing part could call up the next trend of naturalistic stimuli, that is letting subjects have interactions with the environment. Some previous attempts recorded their interpretation (“what came to your mind”, Saalasti et al. Brain Beh 2019) and their recalled memory (“what do you remember”, Baldassano et al., 2017).

We acknowledge this shortcoming briefly as follows and now cite the references provided by this reviewer:

We hereby added a paragraph of this shortcomings in our limitation section:

“In our study, it is important to acknowledge some limitations that need to be addressed. Firstly, the use of the studyforrest dataset poses a limitation due to the fact that the majority of the subjects have already watched the movie and are not English native speakers. As a result, this could potentially introduce a bias into the results obtained, as the comprehension of the movie may not be as accurate as it would be for native speakers. This familiarity with the movie could lead to a diminished surprise level.

Secondly, another limitation of the studyforrest dataset is that it does not allow for cues on individual-level semantic processing itself, which is however a crucial aspect in understanding and interpreting language. The lack of individual feedback restricts our ability to explore further the subjects' response to the movie. This shortcoming could be addressed by extending the studyforrest repository to include the participant's own interpretation of the events in the movie material in a time-point-by-time-point fashion. Indeed, Saalasti and colleagues (2019) recorded subjects' thoughts as a reflection of "what comes into your mind." Additionally, Baldassano and colleagues (2017) asked subjects to recall the movie content to assess "what do you remember."

Saalasti, S., Alho, J., Bar, M., Glerean, E., Honkela, T., Kauppila, M., Sams, M., & Jääskeläinen, I. P. (2019). Inferior parietal lobule and early visual areas support elicitation of individualized meanings during narrative listening. *Brain and Behavior*, 9(5), e01288. <https://doi.org/10.1002/brb3.1288>

5) The methods in places are difficult to follow. I suggest the authors spend some time with for example a colleague who has not seen the manuscript before to spot the places where it is difficult to follow. Increasing clarity throughout would in my opinion help enhance the potential impact of this study.

We have worked on simplifying the methods section.

6) I find it somewhat perplexing that there are only 4 brain states that are discovered. Are these perhaps specific to this movie, which is of Comedy genre?

We thank the reviewer for this helpful point regarding the choice of the number of brain states to be estimated in the HMM model. To address why we have opted for this model selection outcome, we wanted to remind about a fundamental characteristic (which is also one of our main innovations) of our HMM model design, and then clarify our adopted model selection framework across 4 different metrics of model quality (i.e., number of latent factors in the HMM estimation).

In previous HMM modelling works (Vidaurre et al., 2017), due to the frequent constraints regarding brain scanning length in each subject, the mainstream data analysis approach has been to concatenate the brain scans of all subjects together into a single across-subject brain-activity dataframe for downstream steps. The ensuing number of brain states in these common previous analysis designs - typically more than 10 - is therefore shared by the whole group. In contrast, in our present work, we train one specific HMM instance for each of the 15 participants. Further, HMM is known to be intolerant to high-dimensional input data spaces (i.e., less observations in the dataframe are equivalent to relatively more variables to be analyzed per observation). Hence, to reduce the dimensionality, the previous authors would often use PCA or ICA to obtain a reduced version, with less variables to be analyzed (relative to the number of overall brain scans).

As a result, the typically >10 hidden brain states in previous HMM movie modeling studies are also shared across all brain while our 4 brain states are specific to one subject and one region-network combination. From our perspective, our study departed from common earlier analysis designs (especially group wise concatenation) as limitations of previous HMM's modelling and interpretational power. This opportunity came into reach by studyforrest dataset's 2 hour length scan, covering the entire Forrest Gump movie. Intuitively, because our modeling focuses on a rarely analyzed scope, the number of brain states is not identical to that of previous naturalistic neuroscience studies of movies operating at the group level.

From a technical perspective, the optimal number of states is selected via consensus vote across four separate complementary metrics of model fit. In general, there is no single best optimal model or number of latent states for a given model and dataset at hand (Hastie et al., 2001). The goal is to find a model that can adequately describe the brain dynamics. Following this principle, we firstly want to choose the strength of semantic context-brain association as an indicator (Fig. 4 A). i) Specifically, we compared the mean value of the top 10 Pearson's correlation coefficients between HMM models' state presence and 200 extracted semantic contexts for different possible choices of brain state number (x axis). ii) We also modeled subject level variance by summarizing the mean Pearson's correlations across 15 subjects' HMM state presence probabilities (Fig. 4 B). Based on the goal of reflecting idiosyncrasies to the degree possible as supported by the available data, we selected a lower, more conservative value such that more discovered dynamics could be extracted. iii/iv) Lastly, we followed the previous neuroimaging studies and chose a commonly used index (van der Meer et al., 2020), the Bayesian information criterion (BIC). We tested BIC under the condition with limbic subregions and without subcortical subregions (Fig. 4 C and D). BIC measured the likelihood of the model. We prefer a lower value. All these four model selection fidelity metrics searched the range from 2 to 8 candidate states and converge to the conclusion of 4 states. Taken together, across all considered model quality metrics, 4 states was the best available option for our present study.

van der Meer, J. N., Breakspear, M., Chang, L. J., Sonkusare, S., & Cocchi, L. (2020). Movie viewing elicits rich and reliable brain state dynamics. *Nat Commun*, 11(1), 5004. <https://doi.org/10.1038/s41467-020-18717-w>

Vidaurre, D., Smith, S. M., & Woolrich, M. W. (2017). Brain network dynamics are hierarchically organized in time. *Proc Natl Acad Sci U S A*, 114(48), 12827-12832. <https://doi.org/10.1073/pnas.1705120114>

Supplementary Figure 1: Complementary metrics for HMM models converge to 4 state solutions. We justify the selection of 4 HMM states based on four complementary metrics. A) The average value of the top 10 Pearson correlation links between HMM models' state presence and extracted semantic contexts. The larger this value is, the closer the brain activity model aligned with movie narrative features. This value peaked for the choice of 4 states. B) The average Pearson correlations across 15 subjects' HMM state presence. The lower value represented more discovered dynamics. At the point of 4 states, this value decreased rapidly and fluctuated from 4 states to 8 states. C) Bayesian information criterion (BIC) measured the likelihood of the model while penalizing the model complexity. A lower value was preferred. This value reached its global minimum in the range of 3 to 4 states. D) We removed the effect of the limbic partners and evaluated BIC again. This value reached its lowest peak at the point of 4 states.

7) In the illustrations the labels are in places with awfully small font, this should be improved somehow.

We are grateful for this great suggestion and have gone through all our figures with the aim of increasing font size as much as possible.

Reviewer #2 (Remarks to the Author):

The study of Yang et al. extends upon previous work modelling the fMRI signal dynamics elicited by a movie by adopting machine learning to define the semantic context of movie scenes. The approach allowed estimating the likely occurrence of several subject-specific 'region-network combinations' during movie watching that link with information of the evolving narrative (semantic contexts). One of the key findings of this study is the core role of the default mode brain network in tracking and integrating the unfolding storyline. Overall, the paper is clear.

We are deeply appreciative of the reviewer's careful evaluation of our work.

To foreshadow our responses below, we would like to draw attention to a few key elements of our findings that were not emphasized by this reviewer but that we deem core contributions of our present work to the existing literature:

- a) Our work provides a straightforward tool for extracting and interpreting subject-specific dynamic hidden brain states within the context of naturalistic fMRI modeling of movie watching.
- b) To the best of our knowledge, we successfully linked subtitles and movie-flanking narrator information with specific hidden brain states obtained by HMMs. This step forward in subject-specific latent factor modeling was also explicitly endorsed by reviewer 3.
- c) Our study not only investigates the dynamics of major brain networks, but also specifically focuses on as many as 18 anatomical subregions of the amygdala and as many as 38 anatomical subregions of the hippocampus. To the best of our knowledge, this is the first time that neural activity responses, at this fine-grained microstructural granularity, to movies have been studied and perhaps even dynamic fMRI studies in general.

That said, the contribution of this work relative to the existing literature appears incremental. For example, it has been shown that the processing of complex semantic information involves brain regions comprising "slower" high-order networks, particularly the default mode (e.g., works from Hasson and Norman labs). After reading the paper, I was unsure about what new knowledge about brain function was gained.

We appreciate the reviewer's pointing out previous works with earlier conclusions. Indeed, we utilized previous studies as a starting point and motivation for our work and aimed to replicate some similar results and conclusions in the first half of our paper using our new

framework. Nevertheless, our work makes several additional contributions that surpass the current state of the art in naturalistic neuroscience in at least three ways:

- a) Technically, we have developed a framework that shows the feasibility of using hidden Markov models (HMMs) at the single-subject level. The studyforrest resource offers a unique opportunity to this end, which we tried to seize as each subject underwent a 2-hour long brain scan while viewing a movie. By taking advantage of this dataset recording length, we were able to train subject-specific HMM models, which to our knowledge, has not been possible for naturalistic stimuli in the imaging neuroscience community; and we were able to align these subject-specific models across participants. Our ability to estimate subject-specific HMM models allowed us to shed light on important idiosyncrasies of how each individual responds to the movie material - and that information was analyzed on a timepoint-by-timepoint basis. This approach hence provided a comprehensive and faithful picture of the individual's functional coupling dynamics during movie appraisal. The single-subject effects brain map, as for example shown in Figure 3 of our original manuscript, enhances neuroscientific interpretation. Additionally, as our generated subject-specific hidden states differ fundamentally from previous group-wise hidden states, we believe it is both necessary and meaningful to replicate previous work.
- b) Neuropsychologically, we directly compared the value of human-made (52 different variables) and automatically-data-derived (200 different variables) semantic labels in providing insight into neural responses during naturalistic movie watching. We derived a collection of unique semantic contexts from the movie information itself and showed their relevance for tracking specific movie events. These data-driven semantics offered rich descriptions of movie scenes and the evoked neural activity responses, complementing the meaning captured by the human-made labels alone. For example, the semantic context elements that are best explained by each network coupling signature are about various characters and their associated movie events. Hence, this enrichment of information available for quantitative analysis shed new light onto the in-depth functional roles of the default network and hippocampus during movie processing across complementary kinds of rich semantic content.
- c) Anatomically, unprecedented subregion-level granularity was used for the first time for charting and understanding movie responses in functional MRI scans; and perhaps even dynamic functional connectivity analyses in MRI in general. We considered one of 7 major brain network's subregions as defined by the Schaefer Yeo 100 atlas together with 18 amygdala and 38 hippocampal subregions (segmented by taking into account subject specific brain anatomy). We chose not to model brain activity of the entire amygdala or the whole hippocampus, with their functional relationships with the other brain functional coupling patterns. Working at the subregion level gave us the chance to confirm evidence from several known biological pathways that were previously reported in invasive animal experiments. As one example of the fruitful findings from this approach, neurons in the CA1 of the hippocampus (which is one of our 38 hippocampus subregions) have previously been reported to be sensitive to bright vs. dark environments during experiments on light

condition transitions in rodents (Quirk et al., 1990). The same sensitivity to ambient light conditions also emerged in our study in human subjects by using the previously unavailable time-of-day annotation for all movie timepoints provided as part of the studyforrest dataset. Hence, only by working at the subregion level could we build limbic-network interactions and build on evidence from many biological pathways that have been previously reported in other kinds of studies or non-human species.

To clarify our key contributions to the neuroscience literature, we have now added the following summary at the end of the revised introduction section:

“To foreshadow our key contributions, we have developed an analytical framework enabling hidden Markov models (HMMs) at the single-subject level to analyze the dynamic functional connectivity in the brain during naturalistic movie watching. This approach allowed us to understand the idiosyncrasies of each individual's responses to the movie material and provides a comprehensive and faithful picture of the individual's functional coupling dynamics evoked by naturalistic stimulation. Furthermore, we have used unprecedented anatomical granularity at the subregion-level to chart movie responses, mapping out neural responses in 18 amygdala subregions, and in 38 hippocampal subregions. The proposed approach allowed us to build on evidence from biological pathways that have been previously reported in invasive animal experiments. Additionally, we have compared the value of human-made semantic labels and data-driven semantic labels in providing insight into neural responses during movie watching, showing the relevance of data-driven labels to specific movie events and highlighting the broader role of the DN in pooling and binding brain-wide information.”

Moreover, I have several methodological concerns (below) and the significance of the findings is arguably overstated (e.g., delineation of core mechanisms supporting narrative-specific inter-regional coupling).

In the context of the above general impression, I also have the following comments:

1. The authors state that: “our analytical framework was carefully tailored to capture rich information in the full-length 2-hours movie”. It is thus surprising that four hidden brain states represent the optimal resolution, across region-network combinations and subjects. This resolution is coarse (e.g., Vidaurre et al., 2017; van der Meer et al., Nat Comm, 2020) and likely to preclude the detection of meaningful (e.g., region-network or subject-specific) movie-related dynamics.

We thank the reviewer for the helpful point regarding the number of brain states to be estimated in the HMM model. To address why our framework has opted for this number, we first would like to remind us of a fundamental characteristic (which is also one of our main innovations) of our HMM modeling framework and then clarify our methods for principled

selection of the number of latent factors in the HMM estimation (i.e., addressing the model selection problem).

In previous HMM modeling studies (Vidaurre et al., 2017), due to the constraint of typically shorter fMRI scan length available in each subject, the commonly adopted analysis design is to concatenate the brain scans of all subjects - that is, to combine the information from all subjects to one big, aggregate group time series. As a result, previously, the extracted number of hidden brain states (often more than 10) was derived at the aggregate level of the entire group, not at the level of single individuals. Instead, in our work, we could train a subject-specific HMM; one model for each of the 15 participants.

Additionally, the class of HMMs is usually intolerant to high-dimensional input data (i.e., with a lot of variables to be analyzed at the same time). So to reduce the dimensionality of the input space, previous HMM studies commonly used PCA or ICA to obtain a reduced version with fewer input variables to be fitted conjointly by the HMM. However, in our study, as an alternative approach to the curse of dimensionality, we trained an independent HMM model for each of 14 region-network combinations (e.g., the cortical visual network and the subcortical amygdala), which allowed us to perform model fitting on a comfortably much smaller input dimensionality, yielding much more robust model fits. As a result, the 10 or so hidden brain states in previous group-averaged HMM studies were shared across all the brain, while our 4 brain states were specific to one subject and one region-network combination thus yielding $4 * 14 = 56$ states for each subject, being - in total - more states than in previous naturalistic brain-imaging research.

Practically, to address the decision of selecting the number of brain states, we chose the optimal number in a purely data-driven fashion using four separate metrics. In general, usually, there is no single best optimal model or number of latent states for a given dataset (Hastie et al., 2001). The realistic practical goal is usually to find a model that best describes the brain dynamics to the extent supported by the data. Following this principle, we first used the strength of semantic context-brain association as an indicator (Fig. 4A). Specifically, we compared the mean value of the (absolute) top 10 Pearson's correlation coefficients between the HMM models' state presence and 200 extracted semantic contexts with their moment-to-moment expressions. Secondly, we modeled the subject-level variance by summarizing the mean Pearson's correlations across the 15 subjects' HMM state presence probabilities (Fig. 4B). Based on the goal of reflecting idiosyncrasies as much as possible, we selected a lower value so that more dynamics could be extracted. Lastly, to further complement this set of quantitative metrics, we followed a previous study (van der Meer et al., 2020) and used the commonly used Bayesian information criterion (BIC), which measures the likelihood of the model. We tested BIC with and without limbic subregions (Fig. 4C and D), and prefer a lower value. All four charts searched the range from 2 to 8 states and converged to the conclusion of 4 states being the optimal option for our study.

van der Meer, J. N., Breakspear, M., Chang, L. J., Sonkusare, S., & Cocchi, L. (2020). Movie viewing elicits rich and reliable brain state dynamics. *Nat Commun*, 11(1), 5004. <https://doi.org/10.1038/s41467-020-18717-w>

Vidaurre, D., Smith, S. M., & Woolrich, M. W. (2017). Brain network dynamics are hierarchically organized in time. *Proc Natl Acad Sci U S A*, 114(48), 12827-12832.
<https://doi.org/10.1073/pnas.1705120114>

Related to this, I am not convinced that the selection of the optimal number of hidden states should be guided by the correlation between brain activity and movie narrative. This approach may bias the results towards "high order" region-network combinations (see also my comments below). Would it not be more principled to estimate hidden brain states and then assess how such states link to semantic contexts across subjects?

We have taken your suggestion into consideration and have conducted a new quantitative analysis to address the raised point. We used the correlation between brain activity and movie narrative as the index but plotted the different canonical networks separately in different colors. This allowed us to directly evaluate whether the optimal number of hidden states is biased towards "high order" region-network combinations. As shown in the below, different networks are represented by different colors and we observed that the Default, Cont, and DorsAtten networks exhibit peak association strengths at a model complexity of 4 hidden brain states. While the maximum brain-behavior linkage for each number for other major brain networks does not differ much from the number at 4. Both the mean and median values suggest that 4 hidden states are the optimal choice. This suggests that our approach is not biased towards high-order combinations, but rather, reconciles the optimal number of states for all 7 individual canonical networks. We believe that this new analysis strengthens our findings thanks to this valuable reviewer point. Additionally, we have used the commonly used Bayesian information index to determine the optimal number of states, which provides additional support for our decision. Please also note that the number depicted in the Figure 4 is not determined by the Default Network alone, but rather is the mean and median value of all major networks of the widely used Schaefer-Yeo atlas combined.

Additionally, separate hidden Markov model (HMM) models were run for each of the 7 Yeo networks, including the Default Mode Network (DN), as well as all other major brain networks in the Schaefer-Yeo atlas populating the cerebral cortex. We treated all Yeo networks in the same way in our analysis, providing a comprehensive and unique appreciation of all candidate brain network definitions. These additional analyses further strengthen our findings.

Supplementary Figure 12: Optimal number of states selected for seven different canonical networks using the same selection procedure as Supplementary Figure 1 (A).

To determine the optimal number of states for the seven different canonical networks (Supplementary Fig. 12), we used the same selection procedure as described in Supplementary Fig. 1 (A). This involved identifying the number of states that produced the highest average value of the top 10 Pearson correlation links between the Hidden Markov Model (HMM) models' state presence and extracted semantic contexts, which indicated the closest alignment of the brain activity model with movie narrative features.

2. The activity of various region-network combinations during movie watching is likely interlinked. Therefore, I struggle to understand the rationale of only independently assessing brain states in region-network combinations. This approach may obscure critical information on movie-induced whole-brain state dynamics, arguably necessary to interpret region-network-specific fluctuations linked to semantic contexts.

Technically, the application of hidden Markov models (HMM) to high-dimensional input data can be challenging, as a larger number of regions can lead to longer time series data being needed to achieve convergence. Given this limitation, one of two available options are

commonly embraced: a) investigators focus on the entire brain using preliminary compression techniques such as PCA or ICA, or b) investigators focus on a single, canonical network at a time, which is the alternative approach we took in this study. Since the 7 Yeo networks were originally derived from whole-brain functional cluster/matrix decomposition of the PCA/ICA type, these 2 approaches are not as different as they may appear at first.

In other words, previous HMM studies in the field of imaging neuroscience have typically characterized brain responses during movie watching using a single model - a number of high-quality fMRI studies of this kind are already available in the literature. Our study builds upon this work by delving into greater detail, by adopting high-resolution brain atlases of each brain system. In particular, by examining the subregions of the DN and limbic areas, we have been able to reach subregion-level conclusions.

3. The downsampling of text information from 2 seconds to 4 minutes was motivated by a systematic grid. However, the values presented in SFigure 10 are pretty similar between some window lengths. Moreover, the selected window length facilitates the detection of default mode activity (e.g., figure 4 in Baldassano et al., 2017) but may obfuscate the detection of potentially relevant dynamics in “lower” region-network combinations that occur on shorter time scales (e.g., Vis&AM).

Thank you for your feedback and for pointing out the potential role of the downsampling of text information on our results. To respond to this reviewer point, we have now computed a new quantitative analysis. Concretely, we calculated the semantic brain link strength of all brain states with new sets of semantic labels that we have obtained using different window lengths. The average link strength was calculated for both the Default Network and the entire group of 7 networks. The results showed that the average correlation of Default Network was consistently greater than that of the entire group, with the gap between the two increasing as the window length increased. These findings suggest that the DN may play a more prominent role in processing semantic information during movie viewing, relative to other major cortical networks from the Schaefer-Yeo reference atlas.

Supplementary Figure 13: Comparison of average link strength between DN group and the whole group. The figure shows a comparison between the average link strength of the DN group and the entire group. The x-axis represents the window length in seconds and the y-axis represents the correlation. We obtained new sets of semantic labels for different window lengths, and calculated the semantic brain link strength for all of our brain states using these new variables. We then calculated the average link strength for the Default Network (blue line) and for all 7 networks (orange line). The plot provides a visual representation of the difference in average link strength between the DN group and the entire group.

We appreciate your insight and acknowledge that while shorter time scales may reveal potentially relevant dynamics in lower level networks, our focus was on semantics in the present project given the wealth of subtitle and narrator information that is uniquely available in the studyforrest resource. It is important to note that our study is not intended to capture all possible brain dynamics, but rather to identify those that are most strongly associated with the semantic content of the movie. As such, time scales of 5s or 10s would not have been suitable for this purpose. Along with our grid search analysis, as presented in the Supplementary Fig. 13, a window length of 240s is a "sweet spot". This length allows us to identify the difference of semantic correlation between the Default Network and the group. Additionally, this time scale has been shown to have a high correlation with human-rated annotations, which allows our two-pronged approach of data-driven vivid descriptions and human-defined abstract impressions to be maximally effective.

Your concern brings attention to an interesting phenomenon, and we appreciate the opportunity to further consider its implications in our future work. We plan to add the new observation that the gap between Default Network and the whole group's semantic correlation increases with longer window lengths to our content. While we have optimized our method for capturing the most salient semantic associations in the brain, we agree that there may be additional neural activity that is relevant to our understanding of the movie-watching experience but is not captured by our current method. To explore this, future researchers could investigate the feasibility of capturing lower region-network combinations that occur on shorter time scales. Achieving this goal may require a different approach to brain state modeling and a more fine-grained analysis of semantic content.

4. While I can appreciate the general interest in assessing limbic-neocortical activity patterns, the rationale for such analysis is unclear. Why are the amygdala and hippocampus considered key neocortical partners in this context?

The amygdala and hippocampus are crucial components of the subcortical emotion-processing systems in humans. According to the findings presented in the work from Alves and colleagues (2019), the anatomy of the Default Network closely resembles the unitary model of the subcortical emotion-processing systems. Through the coordination of its subregions, the limbic system plays a crucial role in the elaboration of emotions, memories, and different kinds of stimulus-value association - all qualities and capacities that we would deem to be of immediate relevance to movie appraisal. These recent updates on the conceptualization of subcortical emotion-processing systems as a component much closer to the higher association cortex as previously thought further emphasize the importance of studying the relationships between the cortex and the key nodes amygdala and hippocampus to better understand the underlying mechanisms of emotion, memory, and behavior during naturalistic movie appraisal in humans.

In response to this helpful reviewer point, we have extended the relevant paragraph with this rationale:

"In recent years, the conceptualization of the subcortical limbic system as a component closer to the higher association cortex than previously thought has led to increased interest in studying the relationships between the cortex and the limbic key nodes amygdala and hippocampus. These sub-neocortical regions, through their coordination, play a crucial role in the elaboration of emotions, memories, and stimulus-value associations. The anatomy of the Default Network closely resembles the unitary model of the limbic system (Alves et al., 2019), making it an essential component to consider when studying the underlying mechanisms of emotion, memory, and behavior during naturalistic movie appraisal in humans. The interactions among these regions should shed new light on the mechanisms underlying competition for limited computational resources and how the brain captures larger-scale semantic information."

Alves, P. N., Foulon, C., Karolis, V., Bzdok, D., Margulies, D. S., Volle, E., & Thiebaut De Schotten, M. (2019). An improved neuroanatomical model of the default-mode network reconciles

previous neuroimaging and neuropathological findings. *Communications Biology*, 2(1).
<https://doi.org/10.1038/s42003-019-0611-3>

These two brain regions are known to have different levels of association with regions defining canonical brain systems (e.g., activity in the hippocampus has been linked with the activity of core default mode regions, including the angular gyrus and the posterior cingulate). Would these differential functional associations bias the results? If so, how (e.g., increased sensitivity to detect default mode dynamics)?

To clarify, the mentioned analysis was not focused on functional coupling strengths in particular, but on what distributed fMRI signals track most strongly moment-to-moment differences in the evolving movie narrative. As such, our analysis should be sensitive to detect those default mode dynamics that are actually robustly coupled with aspects of semantic cognitive processes in the brain. In contrast, based on the existing corpus of imaging neuroscience literature, hippocampus activity appears not to primarily subserve semantic processing per se, but can contribute or support higher associative semantic processes in the cortical mantle.

In particular, Binder et al. 2009 conducted the largest existing fMRI meta-analysis on various different kinds of semantic reasoning in the brain did not highlight the hippocampus to be involved in semantic processing. These authors state that “using strict inclusion criteria, we analyzed 120 functional neuroimaging studies focusing on semantic processing. Reliable areas of activation in these studies were identified using the activation likelihood estimate (ALE) technique. These activations formed a distinct, left-lateralized network comprised of 7 regions: posterior inferior parietal lobe, middle temporal gyrus, fusiform and parahippocampal gyri, dorsomedial prefrontal cortex, inferior frontal gyrus, ventromedial prefrontal cortex, and posterior cingulate gyrus.” In this quote of core results and in the below figure from their paper, we see that the hippocampus was not identified as a primary region of semantic processing in the brain.

Reviewer #3 (Remarks to the Author):

Summary of the Paper

This paper sought to study the naturalistic responses of humans (15 subjects) to watching the movie *Forrest Gump*, a 2-hour movie. To do this, processed fMRI recordings were paired with natural language descriptions of the movie, namely via a set of augmented subtitles and audio descriptions of the movie. Then, natural language embeddings were constructed from bag-of-words embeddings via a tf-idf transformation followed by non-negative LSA, classical NLP techniques. The result was 200 semantic contexts, which means a matrix $R^{\{T \times 200\}}$, where each of the 200 dimensions corresponds to a different aspect of the movie. On the fMRI side, PCA was applied to the voxels composing the hippocampus and amygdala over time separately, yielding 6-dimensions corresponding to the limbic regions. These were appended to the voxel regions defined by 14 target pre-defined regions of the brain from the Schaefer-Yeo atlas. Then, an HMM model with 4 hidden states with a non-isotropic Gaussian emission model was estimated for each of the 15 subjects and each of the 14 brain regions. For each of the 14x15 combinations, 100 HMMs were fit with different starting seeds and the resultant

HMM whose vector of hidden state probabilities over time (in R^T) best correlated (on average?) over the 200 T-dimensional contexts.

Given this setup, the goal was to better understand the dynamics of brain activity resulting from natural movie stimuli in different subregions of the brain, with special attention to the default mode network (which has been previously shown to have various properties connected with natural stimuli), the hippocampus, and the amygdala.

The paper then associates the estimated latent states of the HMM for each viewer with the 10 most closely related semantic dimensions via correlation analysis, and conducts an analysis of means and variances of time spent in these (now interpretable from a descriptive point of view) hidden states.

Overall, the claimed contributions were as follows:

- Provided a simple method for interpreting the hidden states of naturalistic fMRI data in terms of accompanying narrative text, and conducted studies of the dynamics of various brain regions and their connection w/the text.**
- Lower level parts of the brain like the visual network had fitted hidden states with shorter dwell times and smaller variance**
- Default mode network regions had fitted hidden states with dwell times with higher variance.**
- Different hidden states of the HMM tracked different parts of the narrative, as measured by the semantic context states (e.g., each hidden state correlated with different sparse sets of semantic context dimensions). (Though this doesn't seem surprising since each HMM was chosen to maximize correlation with the semantic context vectors, and the correlations are not that strong ($r \sim .2$)).**
- Functional correlation analyses of the default mode network, and the amygdala demonstrated that anatomically adjacent subregions of both exhibited coupling behavior that was the same in both**
- Another demonstration of the fact that the default mode network appears to capture a lot of the brain activity involved in narrative processing in a variety of ways**
- Dwell times of states varied a lot across subjects**
- Lower level networks had timescales that were less volatile across subjects.**
- Several fine-grained coupling signature pairings for different subregions of the brain.**
- Analysis finding an association between the processing of emotional scenes with the amygdala subregions.**
- Findings of lateralization in the amygdala which correlate with similar lateralizations in neural responses in emotion and work linkage experiments.**

Assessment Summary

Overall, the paper applies simple approaches from machine learning to study naturalistic fMRI narrative data for a long movie. In particular, it appears to be the first work (as far as I can tell) to associate text descriptions with hidden states found via fitting an HMM, and to then conduct a dynamics analysis of the resulting fit.

We are grateful to the reviewer for the thorough evaluation of our paper. We appreciate the positive feedback provided and would like to express our gratitude for the detailed and accurate summary of our work.

The main conclusion seems to be that the default mode network is relevant in narrative processing, but this conclusion has been established in several existing papers already (see survey [1]). In general, I felt that the message of the paper was not clear enough.

We appreciate the reviewer's pointing out previous works with similar conclusions. Indeed, we utilized previous theories as a starting point for our work and aimed to replicate similar results and conclusions in the first half of our paper using our new framework. Technically, we have developed a framework that shows the feasibility of using hidden Markov models (HMMs) at the single-subject level. In the previous HMM modelling works (Vidaurre et al., 2017), due to the constraint of the scan length of each subject, the mainstream analysis design is to concatenate the brain scans of all subjects together. As our generated subject-specific hidden states differ fundamentally from previous group-wise hidden states, we believe it is both necessary and meaningful to replicate previous work.

To clarify the message from our paper, have added this summary of core findings in the introductory section:

“To foreshadow our key contributions, we have developed an analytical framework enabling hidden Markov models (HMMs) at the single-subject level to analyze the dynamic functional connectivity in the brain during naturalistic movie watching. This approach allowed us to understand the idiosyncrasies of each individual's responses to the movie material and provides a comprehensive and faithful picture of the individual's functional coupling dynamics evoked by naturalistic stimulation. Furthermore, we have used unprecedented anatomical granularity at the subregion-level to chart movie responses, mapping out neural responses in 18 amygdala subregions, and in 38 hippocampal subregions. The proposed approach allowed us to build on evidence from biological pathways that have been previously reported in invasive animal experiments. Additionally, we have compared the value of human-made semantic labels and data-driven semantic labels in providing insight into neural responses during movie watching, showing the relevance of data-driven labels to specific movie events and highlighting the broader role of the DN in pooling and binding brain-wide information.”

Vidaurre, D., Smith, S. M., & Woolrich, M. W. (2017). Brain network dynamics are hierarchically organized in time. *Proc Natl Acad Sci U S A*, 114(48), 12827-12832.
<https://doi.org/10.1073/pnas.1705120114>

However, there are several more fine-grained studies which examine dynamics properties and fine-grained function correlations between different subregions of the brain. Also of interest (and novel attention) is the focus on coupling behavior between the default mode network and different parts of the limbic region, the amygdala and the hippocampus, but while there were differences (as elaborated on pg. 14), it was hard to extract clear takeaways and the paper would benefit from more clarity in this direction, particularly in the story of the paper.

Thank you for the reviewer's feedback, we have incorporated their advice and your suggestions to improve the emphasis on our key contributions in the introduction. Our new introduction highlights these three aspects:

1. We proposed a novel approach that uses hidden Markov models (HMMs) to analyze dynamic functional connectivity in the brain during naturalistic movie watching at the single-subject level.
2. We used subregion-level granularity to chart movie responses, which confirmed previous evidence from invasive animal experiments.
3. We compared the relevance of human-made versus data-driven semantic labels in providing insight into neural responses.

We have also taken your feedback into consideration and have decided to sharpen the discussion section for clarity. Our aim is to make the first half of the overall discussion section focused on the replication of previous studies with new methods, and the second half focused on a direct comparison of the DN's coupling behavior with HC and AM. We believe that this approach will provide a clearer and more concise presentation of our findings, and help our readers to better understand the results of our study.

There were also a few issues with the positioning of the paper - there are many highly relevant papers that the literature review missed, which should be added — in particular, many works for a long time have made connections between NLP methods and fMRI data, and sometimes of more sophistication than the methods used here, also on naturalistic, long narrative data (movies, books, etc). So while it does appear that the specific combination of applications of methods is new (HMM to analyze the dynamics of the fMRI data paired with semantic analysis for paired text), it is a stretch to say this is a big innovation. Additionally, the specific methods on each side (the HMM, and the text embeddings) are very simple and have been superseded by other approaches in various cases (see several of the missing papers from the literature review for more recent approaches to applying NLP in the context of fMRI data). On the HMM side, it seems relevant to consider HMM variants that have been specifically tuned and adapted to the problem of event detection in fMRI data [2].

Firstly, we would like to extend our gratitude to the reviewer for bringing to our attention the important literature that was missed in this study. We have taken the necessary steps to include these missing references in the introduction section of our paper to provide a more faithful and comprehensive overview of the field.

Secondly, we are aware of other alternative approaches like general pre-trained large language models. However, these approaches do not offer the same level of fine-grained neuroscientific interpretability as the methods we have used. Our goal was to achieve biologically interpretable results, not necessarily maximize raw prediction accuracy. Although deep learning systems can provide high accuracy, they may not be as transparent in what their results imply for a given application domain such as naturalistic imaging neuroscience (Bzdok & Ioannidis, 2019; Bzdok et al., 2018). Thus, we tried to sweet-spot this balance and achieve a deliberate trade-off between interpretability and accuracy in our approach.

New language models are becoming more powerful and we recognize the potential for comparing AI and the brain as a promising future direction. This idea has been noted in our future considerations. Additionally, we appreciate the suggestion of using HMM variants that are better suited to the task of event segmentation. We plan to include this point in the future direction section as it could provide valuable insights.

“NLP has become an increasingly valuable tool for studying the human language systems implemented in the human brain. Several studies have used advanced NLP techniques to build association maps between language features and brain activity. The diverse set of experimental tasks include predicting movie scenes (Vodrahalli et al., 2018), detecting semantic selectivity (Huth et al., 2016b), and decoding semantic content (Fyshe et al., 2012; Hagoort, 2019). In particular, the usage of NLP techniques to access semantic features of human language has been shown to be successfully mappable to brain activity before (Huth et al., 2016a; Huth et al., 2012). The advent of large language models, such as enabled by transformer architectures, has pointed to exciting features of the human brain, such as the possibility of shared computational design principles in form of next-world prediction mechanisms (Goldstein et al., 2022; Kumar et al., 2022a; Kumar et al., 2022b). Additionally, large open neuroscience datasets have emerged as a promising way to accelerate research in this area (Nastase et al., 2021). These developments are likely to continue to advance new insights into the relationship between language processing and neural processing systems.

”

Bzdok D, Ioannidis JPA. Exploration, Inference, and Prediction in Neuroscience and Biomedicine. *Trends Neurosci.* 2019 Apr;42(4):251-262. doi: 10.1016/j.tins.2019.02.001. Epub 2019 Feb 23. PMID: 30808574.

Bzdok D, Altman N, Krzywinski M. Statistics versus machine learning. *Nat Methods.* 2018 Apr;15(4):233-234. doi: 10.1038/nmeth.4642. Epub 2018 Apr 3. PMID: 30100822; PMCID: PMC6082636.

Tone down the claims of novelty a bit across the whole manuscript

We agree with the reviewer's suggestion to tone down the claims of novelty throughout the manuscript. We will revise the language to better reflect the contributions of our study within the context of the existing literature.

Finally, there were several issues with the presentation - the sentences could be edited for verbosity and clarity, and overall the text would benefit from less purple prose. I also found the descriptions of some of the methodology hard to parse — the paper would benefit from more explicit descriptions with equations, and better notation.

We agree with the reviewer's comments on the presentation of the manuscript and have taken measures to address these raised points. We have simplified sentences, provided more explicit descriptions, and included equations where aspects have not been already published in several previous papers. Additionally, the manuscript has been thoroughly reviewed by native English speakers to ensure that the language used is correct, clear, and concise.

I also thought the connections to understanding consciousness were overreaching.

We appreciate your feedback and concerns about the connections to understanding consciousness.

Regarding your comment, we have taken into consideration the need to strengthen our references to the current reasoning behind consciousness. To address this, we have added more references about Michael Graziano's Attention Schema Theory (cite), which offers a great extent of similarity with our findings of competing for limited computational resources. Additionally, we have also noted that our results are supported by the presence of three different brain regions, each responsible for processing different aspects of sensory information, memory, and emotion. Our discussion of how DN&HC and DN&AM coupling deal with external information and movie events to guide internal (emotion and memory) processing shares a great extent of similarity with Graziano's theory of consciousness. And given limited computational resources, how the brain captures larger-scale semantic information. This further emphasizes the importance of considering limbic pairing in our analysis.

In response to your feedback, we have decided to shorten the paragraph discussing consciousness to better align with the scope of our study. Here is the revised paragraph:

From this:

"Painting a broader canvas to summarize, our analytical framework opened a window to identify and characterize two distinct mechanisms of how the DN dynamically partners with microanatomical subregions of the AM and HC to trace semantically salient changes in the environment, by sifting through a compilation of >20, 000 HMM estimations across seven large-scale networks. In this way, we offer compelling explanations of how some of the deepest brain network layers of the human brain support the active search for meaning and valuable information in the external world – a precondition for judicious choice of candidate actions from the behavioral repertoire (Dohmatob *et al.*, 2020; Hartwigsen *et al.*, 2021). We believe that this capacity may potentially constitute a necessary functional component toward realizing human conscious awareness. Further exploration of these computational design principles can make steps toward solving the riddle of human consciousness (Bengio, 2017)."

To this:

"Painting a broader canvas to summarize, our analytical framework opened a window to identify two distinct mechanisms of how the DN dynamically partners with microanatomical subregions of the AM and HC to trace semantic salience and their changes in the environment, by sifting through a compilation of >20,000 HMM estimations across seven large-scale networks. In this way, we offer explanations of how some of the deepest brain network layers of the human brain support the active search for meaning and valuable information in the external world – a precondition for judicious choice of candidate actions from the behavioral repertoire (Dohmatob *et al.*, 2020; Hartwigsen *et al.*, 2021)."

It could be greatly strengthened by more directly using the NLP components of the analysis to more concretely and strongly analyze the differences between the amygdala+DN and hippocampus+DN combinations. If a more thorough analysis was developed and applied to this part of the paper, then paper would be significantly better. It seems possible to fix this issue without too much additional effort

To address these points head on, we have computed and added the following supplementary analyses:

Supplementary Figure 14: Comparison of PLS model contributions between DN&AM and DN&HC models. (A) We present the comparison of loading parameters for the DN subregions between HC and AM groups using 20-fold cross-validation tests. Z-scored loading values from 20 partial datasets were compared for each subregion using a two-sample t-test. Significance levels are shown on the y-tick marks, with asterisks indicating p-values less than 0.05, 0.01, and 0.001. (B) The same procedure was applied to 200 semantic labels generated by NLP and statistically significant differences were found between the DN&HC and DN&AM groups, as indicated by p-values less than 0.001 for each pair of semantic labels.

Supplementary Figure 16: Lateralization effects of hippocampal and amygdala subregions. Panel A) displays the name of hippocampal subregions on the y-axis, and panel B) displays the name of amygdalar subregions. Each bar represents the mean of the absolute difference in activity across the four brain states, calculated for 15 subjects (4

states for each subject, 60 in total). The error bars indicate the 95% confidence interval, and significance is denoted by *** (p-value < 0.001) for both subregions.

– the conclusion in the last paragraph on pg. 16 is not well supported enough, and the characterization on pg. 17 is too loose and non-rigorous to make stronger claims.

In addition to several new quantitative analyses (cf. above, cf. updated supplementary online material), we have revised as follows.

Last para of page 16:

“Further, our results of cortical functional interplay with limbic partners speak to why and how emotional semantics were tracked by our detected DN&AM signatures. First, the AM has long been treated as the heart of emotion processing in the brain (Bzdok et al., 2013a; Müller et al., 2011). Extending such earlier findings to subregion granularity, we now brought to the surface the complementary lateralization effects from AM activity. The subregions with stronger contributions in the left-hemispheric amygdala usually showed weaker functional contributions in the right hemispheric amygdala. Conversely, the amygdala subregions with weaker contributions in the left hemisphere tended to play stronger roles in their counterparts in the right hemisphere. Similar asymmetric effects of neural responses were also discovered in a previous emotion and word linkage experiment from electrophysiological recordings in humans (Abbassi et al., 2011). Therefore, our lateralization findings in the AM further confirmed and explained how emotionally evocative semantic contexts are tied to the subregion-specific lateralization effects in the AM.”

Characterization on pg.17:

“Second, coherent with recent reasonings (Bzdok et al., 2015; Dohmatob et al., 2020), our discovered movie-induced coupling interactions between the PMC and specific AM subregions dovetail with their putative implications in external environment monitoring, especially significance detection and self-relevance evaluation. Moreover, the derived external descriptions offered rich contextualization for neuroscientific interpretation of the extracted limbic-neocortical interaction patterns that appear to directly speak to the attention deployment theory of emotion control (Ferri et al., 2016; Nummenmaa et al., 2012; Ochsner et al., 2012). That is, our findings may reflect attention reallocation mechanisms that came to bear when subjects were viewing unpleasant movie scenes (van der Meer et al., 2020). For example, “valence” became apparent as one of the leading annotations in three out of four DN&AM signatures: Indeed, the functional coupling signatures associated with negative valence annotations related to more unpleasant semantic scenes of the movie plot (e.g., war). In the face of complex affective semantics in a real-world simulation experiment, we thus linked adaptive emotion regulation with flanking functional coupling changes between the highly associative PMC and dedicated AM circuits.

More broadly, our collective findings motivate an extension to the traditional AM survival theory by means of higher semantic reflection. According to the classical view (LeDoux,

2012), humans show intuitive responses to sudden changes in the ambient environment. For example, if a person saw a bear chasing, their adrenaline level would surge automatically, as an instance of a fight-or-flight reaction mediated by the sympathetic nervous system. Revising this classic notion, the AM survival theory may benefit from integration with neural processes subserved by the recently evolved deepest neural processing layers: based on continuous conscious awareness of consistency or discrepancies of environmental features, the DN may potentially liaise with dedicated AM subregions to support scanning the external world for self-relevant information and otherwise emotionally evocative cues. In this way, emotionally edited sensory information can be instrumental to the higher association circuits by giving color to a vast number of candidate semantic interpretations and by effectively directing the allocation of attentional resources based on an evaluation system of significance for the organism (Raymond, 2009; Schupp et al., 2007; Taylor and Fragopanagos, 2005). After detecting behaviorally relevant information in the environment, the human brain also needs to integrate semantic knowledge into the memory system to store information and compare it against past experiences to help with upcoming decisions on how to act on the world.

Across the delineated DN&HC signatures, medial and lateral parts of the PFC showed functional coupling dynamics with designated HC subregions, as another core limbic partner of the higher association cortex. HC-PFC pathways have been discussed before to be involved in episodic scene construction and memory (Eichenbaum, 2017). According to previous reflections, the PFC is implicated in the suppression of content-independent stimuli to boost information retrieval from the environment. Instead, the HC probably subserves retrieval and organization of content-related memory (Eichenbaum, 2017). Our HC subregion-level delineation showed that neural responses of CA1-4 and subiculum (especially the head segments) were functionally interlocked with PFC activity responses. This observation confirms and details previous reports on the HC-PFC pathway: it is well-established that the PFC receives direct axonal projections from the hippocampal subiculum and CA1 in both animals (Aggleton et al., 2015; Barbas, 1995; Carmichael and Price, 1995) and humans (Riley and Constantinidis, 2016; Wael et al., 2018). In these studies, the medial PFC, as opposed to its lateral parts, was typically more emphasized for its dense fiber bundle connections to the HC subregions that our quantitative analyses here spotlight during movie engagement (Vertes et al., 2007)."

It would also benefit the paper to highlight this directional point in the beginning of the paper, rather than focusing on how NLP has not been applied to long narratives before (which is false), and also to avoid the emphasis on the finding that the default mode network plays a very strong role in narrative processing, which is not a novel finding (though it is always nice to see more evidence).

We have toned down the aspects mentioned by this reviewer.

To clarify our key contributions to the neuroscience literature, we have now added the following summary at the end of the revised introduction section:

"To summarize our key contributions, we have developed a new framework using hidden Markov models (HMMs) at the single-subject level to analyze the dynamic functional connectivity in the brain during naturalistic movie watching. This approach allows us to understand the idiosyncrasies of each individual's responses to the movie material and provides a comprehensive and faithful picture of the individual's functional coupling dynamics evoked by naturalistic stimulation. Furthermore, we have used unprecedented subregion-level granularity to chart movie responses, considering 18 amygdala subregions, and 38 hippocampal subregions. The proposed approach allowed us to confirm evidence from biological pathways that have been previously reported in invasive animal experiments. Additionally, we have compared the value of human-made semantic labels and data-driven semantic labels in providing insight into neural responses during movie watching, showing the relevance of data-driven labels to specific movie events and highlighting the broader role of the DN in pooling and binding brain-wide information."

[1]: The default mode network: where the idiosyncratic self meets the shared social world. Yaara Yeshurun, Mai Nguyen, and Uri Hasson

[2]: Discovering event structure in continuous narrative perception and memory. C. Baldassano, J. Chen, A. Zadbood, J.W. Pillow, U. Hasson, K.A. Norman.

Some Example Missing Citations

[3]: Narrative event segmentation in the cortical reservoir. Peter Ford Dominey

[4]: Mapping neural activity to language meaning. L. Wehbe, A. Fyshe, T. Mitchell (and many, many previous works by these authors)

[5]: Aligning context-based statistical models of language with brain activity during reading. L. Wehbe, A. Vaswani, K. Knight, T. Mitchell

[6]: Mapping between fMRI responses to movies and their natural language annotations. Kiran Vodrahalli, Po-Hsuan Chen, Yingyu Liang, Christopher Baldassano, Janice Chen, Christopher Honey, Uri Hasson, Peter Ramadge, Kenneth A. Norman, Sanjeev Arora.

[7]: Decoding the Semantic Content of Natural Movies from Human Brain Activity. Alexander G. Huth, Tyler Lee, Shinji Nishimoto, Natalia Y. Bilenko, An T. Vu, Jack L. Gallant. (And many many previous works by these authors).

[8]: Low-dimensional Structure in the Space of Language Representations is Reflected in Brain Responses. Richard Antonello, Javier S. Turek, Vy Vo, Alexander Huth.

[9]: Bayesian Surprise Predicts Human Event Segmentation in Story Listening. Manoj Kumar, Ariel Goldstein, Sebastian Michelmann, Jeffrey M. Zacks, Uri Hasson, Kenneth A. Norman

[10]: Reconstructing the cascade of language processing in the brain using the internal computations of a transformer-based language model. Sreejan Kumar, Theodore R. Sumers, Takateru Yamakoshi, Ariel Goldstein, Uri Hasson, Kenneth A. Norman, Thomas L. Griffiths, Robert D. Hawkins, Samuel A. Nastase.

[11]: Narratives: fMRI data for evaluating models of naturalistic language comprehension

Nastase, S.A., Liu, Y., Hillman, H., Zadbood, A., Hasenfratz, L., Keshavarzian, N., Chen, J., Honey, C.J., Yeshurun, Y., Regev, M., Nguyen, M., Chang, C.H., Baldassano, C., Lositsky, O., Simony, E., Chow, M.A., Leong, Y.C., Brooks, P.P., Micciche, E., Choe, G., Goldstein, A., Vanderwal, T., Halchenko, Y.O., Norman, K.A., & Hasson, U.

Questions

1. How exactly do you label the 200 semantic dimensions with names beyond looking at word clouds?

We treated word clouds as the names of our semantic labels. We provided vivid descriptions of the movie events that are most closely linked to this semantic label by assigning one sentence description, in order to aid readers who may not be familiar with the movie's contents.

How did you associate them with emotions?

The data-driven semantic labels and human-curated annotations used in the study are mutually supportive. They both serve as external descriptors to correlate and explain brain dynamics, but they differ in their focus. Semantic labels are detailed and focus on specific events in the movie, while annotations are more abstract and provide a broader understanding of the emotional, locations, etc. Despite their differences, the two types of labels complement each other and are used in parallel to gain a more comprehensive understanding of the brain's response to naturalistic stimuli. For example, when a certain brain state has high correlations with a semantic context and an emotional entry, we would explain the brain states with these two contexts.

I didn't quite understand the part about the 52 annotations. (In particular, what exactly is being annotated? The word clouds from the semantic vectors? The movie scenes? The text of the movie? The reactions of the viewers?)

The 52 annotations were manually obtained from the studyforrest dataset to describe the content of the movie (original movie scenes). Specifically, nine female observers, who were all students, participated in annotating emotional entries for the audio-visual version of the movie (Labs et al., 2015). None of the observers had participated in a previous brain-imaging study. The movie was divided into 205 cinematographic scenes and each observer was asked to annotate the emotional expressions present in a random order to reduce

carryover effects. For each expression of emotion, the observers noted the start and end time in seconds as well as the specific emotion expressed. Other annotations include the physical location of each scene (e.g., D.C.), whether it took place indoors or outdoors, and whether it occurred during the day or at night. The movie scenes were also annotated by two individuals, one of whom had a background in filmmaking. In short words, the 52 annotations were human-curated and offered by the studyforrest dataset.

Labs, A., Reich, T., Schulenburg, H., Boennen, M., Mareike, G., Golz, M., Hartigs, B., Hoffmann, N., Keil, S., & Perlow, M. (2015). Portrayed emotions in the movie "Forrest Gump". F1000Research, 4.

2. How does your HMM fit compare to the event segmentation HMM of Baldassano et. al. [2]?

Our work focuses on the subject level, prioritizing individual differences or idiosyncrasies, whereas the event segmentation HMM of Baldassano and colleagues (2017) is based on a group average approach which prioritizes accuracy and generalizability of event segmentation. As a result, their segmented movie events are based on the whole group while ours are based on a single subject.

Additionally, our HMM's observation model is different from theirs. Specifically, their HMM uses an isotropic Gaussian observation model, which is a simplified version of multivariate Gaussian. Because of our focus on one single region-network combination and 2-h length scan time, we can afford the complexity of the original multivariate Gaussian observation model.

This approach allows us to better address idiosyncrasies at the individual level. We believe that our approach is more appropriate for our research question, while the group average approach may be more appropriate for others. It is important to carefully consider the specific research question and data characteristics when choosing the appropriate approach for event segmentation using HMMs.

3. How exactly do you calculate dwell times from the estimated probabilities of states? Do you pick the max probability per time point and assign that state as the label? I missed it if this was stated somewhere.

To calculate the dwell times from the estimated probabilities of states, we followed the same approach as in previous papers. To answer your specific question, yes, we picked the maximum probability per time point and assigned that state as the label, as per the nature of the HMM. One of the key properties of the HMM is that it yields both the all-or-nothing assignment of each time point to one of the latent brain states and the probabilities of the extent to which each brain state is reflective of each time point (cited Bishop's book). We revised the according methods section to provide more detailed explanations on this part in our manuscript:

"To analyze the temporal characteristics of our derived brain states in greater detail, we calculated the dwell times of each HMM state, which refers to the duration of time that a given state is visited. To compute the dwell times, we adopted the same method used in previous studies (Vidaurre et al., 2017). Specifically, we assigned each time point to the state with the highest probability and calculated the duration of time that each state was visited. We then aggregated the dwell times at the single subject level by comparing the average and volatility across four states for the same HMM model. This allowed us to compare the processing timescale differences across 14 region-network combinations. Following this, we averaged the temporal characteristics across 210 estimated HMM instances (15 subjects x 14 region-network combinations). In doing so, we were able to directly characterize how the temporal processing characteristics varied across subjects and functional network level dynamics."

Vidaurre, D., Smith, S. M., & Woolrich, M. W. (2017). Brain network dynamics are hierarchically organized in time. *Proc Natl Acad Sci U S A*, 114(48), 12827-12832.
<https://doi.org/10.1073/pnas.1705120114>

4. If you use the full trajectory of semantic contexts and HMM fits to choose the best HMM fit, you are using information from the whole movie. This is a problem when you do your held-out regression analysis with partial least squares, since the fitting of the unsupervised step uses information from the whole movie.

Thank you for raising this important question about our method. We understand your point about the use of information from the whole movie in the held-out regression analysis.

We would like to clarify that, the HMM model does not lend itself to a straightforward cross-validation scheme. Our approach follows the guideline mentioned in the Hastie et al. (2001, *Elements of Statistical Learning*), which states that:

"In general, with a multistep modeling procedure, cross-validation must be applied to the entire sequence of modeling steps. In particular, samples must be "left out" before any selection or filtering steps are applied. There is one qualification: initial unsupervised screening steps can be done before samples are left out. "

5. On pg. 9, are the default mode network regions anatomically adjacent to each other? What is the relation to the amygdala subregions? The language is not clear.

Thank you for your question. In Figure 3D, we used the Schaefer Yeo 100 atlas to label the default mode network regions. The anatomically adjacent regions would be labeled under the same name but with different numbers, such as LH-DN-PFC-1 and LH-DN-PFC-2. However, the DN region is made up of different separate regions, such as LH-DN-PFC and LH-DN-Temp, so these regions are not necessarily adjacent to each other.

Regarding the amygdala subregions, we derived six principal components from the subregion-level data, with three for each hemisphere. These principal components are represented by the last six entries (bottom six rows or columns) in Figure 3D. We apologize for any confusion caused by our previous language, and we have updated the relevant part for clarity:

"Figure 3. Raters' and data-driven descriptions of story events show complementary links with brain state dynamics across a 2-hour movie. We supplemented our semantics-brain links with external-rater-curated annotations based on traditional neurocognitive concepts. A) The top 10 correlated semantic contexts with brain state 1 of subject 1's DN&AM model (HMM). We highlighted the projected embeddings of our exemplary semantics context (No. 152) in pink. B) Pearson's correlation between hand-made annotations and the semantic context No. 152. "Happiness", "positive valence", and "Gump property" were closely linked with the embedded context features (for the other 199 semantic contexts, see Supplementary Fig. 2). C) Brain renderings show DN&AM region-network contributions from dynamic brain state 1 (for the other states, see Supplementary Fig. 3). The most prominent subregion was the right middle temporal gyrus in the right hemisphere. D) Functional coupling links among DN and AM subregions of state 1. **The DN's subregion names are from Schaefer Yeo 100 atlas, where the bottom / rightmost 6 rows are principal components for the amygdala's left and right hemispheres.** The left and right hemispheres of the AM related to DN in different patterns (for the other brain states, see Supplementary Fig. 4). "

6. It would be good to quantify lateralization effects rather than describe them; I felt the description was imprecise and do not amount to an argument for explaining "links between the brain and movie contextual information" - at least I don't see how this conclusion follows.

We thank the reviewer's advice, and we added three follow-up quantitative analyses to quantify the lateralization effects of DN networks, amygdala subregions, and hippocampus subregions.

To investigate lateralization effects in the Default network, we were unable to perform subregion-level comparisons due to the asymmetric segmentation in the Schaefer Yeo atlas. Therefore, we utilized the mean activity defined by a multivariate Gaussian model for each of the four brain states and calculated its mean absolute value across subregions separately for the left and right hemispheres. The results are summarized in Supplementary Fig. 15, where each bar represents a unique brain state and is grouped by amygdala and hippocampus pairings. The solid bars represent the left hemisphere while the transparent bars represent the right hemisphere. Our findings indicate a significant lateralization effect, as the left Default network activity is significantly smaller than that of the right hemisphere.

Supplementary Figure 15. Lateralization effects of the Default network. In the bar plot, we show the mean absolute activity across left and right hemisphere Default network (DN) subregions. The solid bars represent the left hemisphere, while the transparent ones represent the right hemisphere. Across four brain states and different subcortical partners, we consistently observe smaller activity in the left hemisphere compared to the right hemisphere. We performed a two-sample t-test to test for group differences, and the results were significant ($p\text{-value} < 0.001$).

Then, because our structural data defined amygdala and hippocampus segmentations are symmetrical, we performed the subregion-level detection (Supplementary Fig. 16). In order to detect subregion-level lateralization effects, we used the mean activity defined by a multivariate Gaussian model of each brain state. We employed each subject's original PCA model to inverse-transform the principal components back to subregion-level plotting. It should be noted that the left and right principal components share the same PCA model, which eliminates the variance caused by model training. We calculated the absolute difference between left and right regions and performed a one-sample t-test to test if the group of absolute difference values was greater than zero. All results were significant.

Notably, we also found that the trend of lateralization varied across subjects. Specifically, for some subjects, the left subregions showed greater activity than the right, while for others, the opposite was true. We want to thank the reviewer for their insightful comments, which led us to this new observation.

Supplementary Figure 16: Lateralization effects of hippocampal and amygdala subregions. Panel A) displays the name of hippocampal subregions on the y-axis, and panel B) displays the name of amygdalar subregions. Each bar represents the mean of the absolute difference in activity across the four brain states, calculated for 15 subjects (4

states for each subject, 60 in total). The error bars indicate the 95% confidence interval, and significance is denoted by *** (p -value < 0.001) for both subregions.

7. In Fig. 4, could you include a pointer to the section of Methods you're referring to? Also, it's not clear to me why the box plots in Fig. 4 for regions other than the limbic region are not identical across the amygdala and hippocampus — I didn't completely understand the methods section description of what Fig. 4 means. Is it because of the fact that you concatenate either the amygdala or hippocampus 3-dim PCA embedding to the brain regions, and this affects the fit of the HMM? If so this wasn't clear from the description.

We apologize for any confusion caused by the naming of the limbic regions in our paper. The limbic regions colored in cream in Figure 4 are part of the cortical network provided by the Schaefer Yeo atlas, and are not the same as the ROIs we derived from the hippocampus and amygdala subregions. As a result, it is reasonable to expect that these different sets of ROIs would not generate identical results.

To improve clarity, we decided to change names for our hippocampus and amygdala. We have made the following revision to the manuscript (results section):

"Henceforth, we refer to the amygdala and hippocampus as *limbic, non-neocortical or subcortical* structures, as they are not included in our cortical Schaefer-Yeo atlas. However, it should be noted that the hippocampus is part of the allocortex and therefore sometimes considered as subcortical and sometimes as a cortical region."

To clarify this point, we have revised the method description and added a pointer to the method section in Figure 4. Here is the updated version of the Figure 4 and the related description:

"Figure 4. Functional coupling dynamics in the deepest brain network layers show the most intimate relation to the unfolding movie narrative. To compare the strength of semantics-brain links from lower (visual network, Vis) to higher network (default network, DN) layers in the brain, we computed the average Pearson's correlation strength between semantic context expressions and the presence of dynamic brain states in each of 15 subjects (cf. "Subject-level association analysis between movie events and dynamic brain states" section in Methods). Each color denotes one of the seven canonical functional networks (according to the Schaefer-Yeo atlas definition). The collective amygdala (left) or hippocampus subregions of the limbic system were jointly analyzed with these intrinsic functional networks (cf. methods). The DN showed the most prominent median value, relative to the six other brain networks, which indicated this highly associative neural processing layer as most dominant in consistently tracking the semantic richness in the evolving movie narrative. Top right: an exemplary permutation test of the DN&AM model. The red dash lines show the mean semantics-brain link strengths of 15 subjects, and the blue bars show the null distribution by shuffling the state presence timeseries 1000 times. Boxplot: upper (lower) edge of the box is the 25th (75th) percentile (interquartile distance);

the middle line is the median value; the green triangle shows the mean value; the whiskers summarize the extreme data points of the distribution of median semantics-brain associations.”

8. How similar are the 4 “brain-states” (hidden states of the fit HMM) across the 210 total HMMs? It’s not clear that these should be similar, right?

Thank you for your response. To clarify, it seems that you are questioning whether the four hidden brain states we identified using the HMM approach should be expected to be similar across the 14 region combinations and 15 subregions used in our analysis. We agree that the HMM approach is non-convex, and therefore the resulting brain states do not necessarily need to be similar. This is why we have estimated an excess of HMM for each subject and each region-network combination to identify the HMM instances that best match at the group level.

In addition, we used different numbers of ROIs for different region combinations, and therefore it would not be reasonable to expect the resulting brain states to be similar across these combinations. Finally, as we demonstrated in Figure 5 and the bar plot range in Figure 4, the brain states varied significantly across different subjects.

9. How do you account for the variation in the HMM fit? It seems to me that it’s plausible that after you pick the HMM fit with maximal correlation to the semantic contexts over 100 initializations, the fact that you are still doing 210 HMM fits (and then average over 14 of them per person) suggests that there could be significant variance in the estimated states due to the fitting process and thus the estimated dwell time - it’s not clear that this is only due to the subjects being different. Can you do a statistical test over the variation of the HMM fitting process to check the significance of the standard deviation variations over each human ?

To recap, our goal in training the HMM models was to generate 210 unique models (each for each of 15 subjects x for each of 14 different region network combinations) for post hoc analysis. For each of these 210 models, we ran 100 different models using a random initialization method of model parameter values because HMM is a non-convex model. After obtaining these 100 candidate HMM model estimates in each context, we selected the one among the 100 that offered the highest correlation and discarded the other 99 models. As a result, we cannot perform a statistical test between the selected optimal HMM instance and the other 99 other estimates of model instances.

10. Fig. 6 and Fig. 7 would be much clearer to understand if you wrote down the partial least squares objective.

Thank you for your suggestion. We agree that it would be helpful to include the partial least squares (PLS) objective in Figs. 6 and 7 to improve clarity. However, the objective has been

clearly stated in the methods section. We decided not to update the figures to include the objective for the sake of brevity, since this tool has been used in the neuroimaging community for 25 years.

11. For the results on pg. 14, it would be nice to have a quantitative summary of the associations, at least for the main points you want to make (e.g. amygdala and emotions).

Thank you for your suggestion. We agree that including a quantitative summary of the associations in our results would enhance the clarity of our findings. Based on your previous suggestions, the Supplementary Figure 14 has been created to serve this purpose.

Supplementary Figure 14: Comparison of PLS model contributions between DN&AM and DN&HC models. (A) We present the comparison of loading parameters for the DN subregions between HC and AM groups using 20-fold cross-validation tests. Z-scored loading values from 20 partial datasets were compared for each subregion using a two-

sample t-test. Significance levels are shown on the y-tick marks, with asterisks indicating p-values less than 0.05, 0.01, and 0.001. (B) The same procedure was applied to 200 semantic labels generated by NLP and statistically significant differences were found between the DN&HC and DN&AM groups, as indicated by p-values less than 0.001 for each pair of semantic labels.

This is similar to how we have reported quantitative summaries of associations of multivariate models repeatedly in our previous research:

Poeppel, T. B., Dimas, E., Sakreida, K., Kernbach, J. M., Markello, R. D., Schöffski, O., ... & Bzdok, D. (2022). Pattern learning reveals brain asymmetry to be linked to socioeconomic status. *Cerebral Cortex Communications*, 3(2), tgac020.

Style Issues and Grammar etc.

*** Equations should be clearly written out. It would be helpful to be clearer about the dimensions of various matrices and vectors, should explicitly add the optimization problem formulations, etc.**

We agree with this suggestion very much in general. In the present scenario, we have used well-known models in novel ways. As such, the equations of the entertained quantitative models, such as HMM and PLS, are exactly the same as in hundreds of other neuroscience papers. It is in how we have designed the feature spaces and how we have brought to bear these models to attack previous research questions in naturalistic neuroscience in alternative ways.

**** For instance, on pg. 13, the description "extract the dominant signature that tracks how the 200 semantic context expressions..." is very confusing to read. What is a "signature?" It would be much better if the PLS-R objective was just written out and the goal more clearly described.**

A 'signature' refers to one of the extracted latent components (=latent factors) of the PLS model. These are sorted by nature, hence the 'leading' signature refers to the first PLS latent component, analogous to previous PLS papers.

*** As an example of word choice issues - on pg. 13, "triangulated" is an odd choice of word to use - maybe "associated".**

We have followed this great suggestion and replaced by 'associated'.

*** It would be better to avoid using "*" for denoting matrix shapes, better is "x" or "\times".**

Ok, done.

*** There are some odd tense choices throughout (past tense a lot)**

Using mostly past tense in methods and results section is a standard for all publications from our lab.

*** There is a lot of passive voice that could be removed.**

We have carefully gone through the entire manuscript to rephrase passive voice into active voice sentences.

*** There is a lot of unnecessary verbiage and unnecessary adjectives.**

We have diligently worked on our manuscript draft to simplify the language and overall presentation of our findings and conclusions.

Reviewer #1 (Remarks to the Author):

The authors have in my opinion satisfactorily addressed all my concerns and suggestions

Reviewer #2 (Remarks to the Author):

Excellent rebuttal. Thank you for highlighting the novelty of this work and addressing my additional concerns.

Reviewer #1 (Remarks to the Author):

The authors have in my opinion satisfactorily addressed all my concerns and suggestions

We sincerely appreciate your positive feedback and thank you for acknowledging that we've satisfactorily addressed your concerns and suggestions.

Reviewer #2 (Remarks to the Author):

Excellent rebuttal. Thank you for highlighting the novelty of this work and addressing my additional concerns.

We sincerely appreciate your positive feedback and thank you for acknowledging that we've satisfactorily addressed your concerns and suggestions.